# Enhanced β-adrenergic signalling underlies an age-dependent beneficial metabolic effect of PI3K p110α inactivation in adipose tissue

Caroline Araiz[1], Anqi Yan[1], Lucia Bettedi[1,5], Isabella Samuelson [1,6], Sam Virtue[2], Anne K. McGavigan[2], Christian Dani[4], Antonio Vidal-Puig[2,3] & Lazaros C. Foukas [1]

The insulin/IGF-1 signalling pathway is a key regulator of metabolism and the rate of ageing. We previously documented that systemic inactivation of phosphoinositide 3-kinase (PI3K) p110α, the principal PI3K isoform that positively regulates insulin signalling, results in a beneficial metabolic effect in aged mice. Here we demonstrate that deletion of p110α specifically in the adipose tissue leads to less fat accumulation over a significant part of adult life and allows the maintenance of normal glucose tolerance despite insulin resistance. This effect of p110α inactivation is due to a potentiating effect on β-adrenergic signalling, which leads to increased catecholamine-induced energy expenditure in the adipose tissue. Our findings provide a paradigm of how partial inactivation of an essential component of the insulin signalling pathway can have an overall beneficial metabolic effect and suggest that PI3K inhibition could potentiate the effect of β-adrenergic agonists in the treatment of obesity and its associated comorbidities.

[1] Institute of Healthy Ageing & Department of Genetics, Evolution and Environment, University College London, London WC1E 6BT, UK. [2] University of Cambridge Metabolic Research Laboratories, Wellcome Trust-MRC Institute of Metabolic Science, Addenbrooke's Hospital, Cambridge CB2 0QQ, UK. [3] Wellcome Trust Sanger Institute, Hinxton CB10 1SA, UK. [4] Université Côte d'Azur, CNRS, Inserm, iBV, Faculté de Médecine, 06107 Nice Cedex 2, France. [5] Present address: National Institutes of Child Health and Human Development (NICHD), Bethesda, MD 20814, USA. [6] Present address: University of Cambridge Metabolic Research Laboratories, Wellcome Trust-MRC Institute of Metabolic Science, Addenbrooke's Hospital, Cambridge CB2 0QQ, UK. These authors contributed equally: Caroline Araiz, Anqi Yan. Correspondence and requests for materials should be addressed to L.C.F. (email: l.foukas@ucl.ac.uk)

nterventions to increase the lifespan and healthspan of organisms have been the subject of intense research efforts. Loss-of-function mutations in components of the insulin/IGF-1 signalling (IIS) pathway have long been known to have a life- and healthspan extending effect in diverse model organisms[1]. The fact that inactivating mutations in a pathway essential for regulation of metabolic physiology extend lifespan is seemingly paradoxical and has been referred to as the insulin paradox[2]. Although evolutionary theories have been evoked to explain the phenomenon of lifespan extension through loss-of-function mutations in essential genetic pathways such as the IIS, mechanistic insights as to how this is achieved at the molecular level have only recently started to emerge[3,4].

A number of genetic and other interventions that extend lifespan/healthspan have also beneficial effects on metabolic physiology, highlighting the interconnectedness of metabolism and ageing[5]. Prime examples of gene mutations that extend lifespan and improve glucose and lipid homoeostasis in old mice are mutations in the growth hormone/IGF-1 signalling axis in hypopituitary dwarf long-lived mutant mice[6]. Furthermore, calorie restriction, the best known intervention to extend lifespan in a number of species, has also profound beneficial metabolic effects, manifested as improved insulin sensitivity and glucose homoeostasis in calorie restricted mice[7].

In the context of interventions shown to affect both lifespan and metabolism, the adipose tissue has a prominent role. The adipose tissue is an important organ in the regulation of energy homoeostasis[8]. Moreover, the adipose tissue has been shown to play an important role in the lifespan-extending effects of insulin pathway downregulation. A seminal finding in this regard has been the demonstration that adipose tissue-specific insulin receptor knockout mice (FIRKO) display extended longevity and they are also protected against age-related metabolic pathologies[9].

We and others have documented that p110α is the key PI3K isoform in the insulin signalling pathway[10,11]. We recently reported that global partial inactivation of p110α has an age-dependent beneficial effect on metabolism and causes a modest increase in lifespan[12]. To gain further insights into the mechanism underlying these effects, we assessed the role of the adipose tissue in the beneficial metabolic impact seen upon global p110α inactivation. To this end, we inactivated p110α specifically in the mouse adipose tissue by Cre/loxP approaches and characterised the metabolic profiles of these mice over the course of ageing, which is one of the most physiologically relevant stresses an experimental model can be subjected to. We report here that partial inactivation of p110α potentiates β-adrenergic signalling in the adipose tissue. This results in enhanced catecholamine-induced energy expenditure (EE), which counteracts the effects of insulin resistance thus preserving normal glucose homoeostasis over the course of ageing. These findings have clear implications for the development of therapeutic interventions for age-related obesity and associated comorbidities.

## Results

**p110α deletion in adipose tissue causes insulin resistance.** We applied Cre/loxP-mediated conditional gene deletion to inactivate p110α in the adipose tissue of mice by crossing conditionally targeted p110α mice (p110α$^{FLOX}$)[13] with a transgenic line expressing Cre under control of the adiponectin promoter (Adipoq–Cre)[14], which is more specific for deletion in the adipose tissue than the earlier reported aP2–Cre line[15]. We isolated metabolic tissues from mice expressing the Adipoq–Cre transgene (p110α$^{DEL}$) or not (p110α$^{FLOX}$), and assessed p110α protein expression by immunoblot analysis. Significant reduction of p110α protein expression was only seen in the white and brown

adipose tissues (WAT and BAT) of p110α$^{DEL}$ mice, with a reduction of an average of 70% (WAT) and 90% (BAT), compared to the expression levels in p110α$^{FLOX}$ littermates (Fig. 1a, b and Supplementary Fig. 1). The expression of p110β, the other broadly expressed PI3K isoform, as well as that of the p85 regulatory subunit was not affected under these conditions in p110α$^{DEL}$ mice.

We next tested insulin sensitivity in metabolic tissues of approximately 1-year-old mice by injecting insulin and assessing PI3K activity and Akt phosphorylation as readouts of insulin pathway activation. Consistent with the previously demonstrated role of p110α in insulin receptor signalling[10,11], the PI3K activity associated with IRS-1 immunoprecipitates was significantly reduced in visceral (epididymal) WAT (eWAT) of p110α$^{DEL}$ mice (Fig. 1c). In line with this, insulin-induced Akt phosphorylation was also significantly impaired in eWAT, but only marginally affected in iWAT and BAT isolated from p110α$^{DEL}$ mice (Fig. 1d). Akt phosphorylation was impaired to a lower extent also in the liver and skeletal muscle of p110α$^{DEL}$ mice (Supplementary Fig. 2), indicating the development of systemic insulin resistance. This was further corroborated by hyperinsulinemic–euglycemic clamp experiments in 20-week-old mice that revealed profound systemic insulin resistance at this age point, as evidenced by an impaired rate of glucose disappearance, increased hepatic glucose production and elevated free fatty acids in the plasma in p110α$^{DEL}$ mice (Fig. 1e) under clamp conditions. Notably, the rate of glucose disappearance at basal level was slightly elevated in p110α$^{DEL}$ mice, which suggests that p110α$^{DEL}$ mice may have a higher basal metabolic rate under clamp conditions.

**p110α$^{DEL}$ mice maintain normal glucose tolerance over ageing.** We assessed the metabolic profile of p110α$^{DEL}$ mice by performing glucose tolerance and insulin tolerance tests over a number of time points ranging from 3 to 24 months of age. In line with the impaired insulin signalling in metabolic tissues and hyperinsulinemic–euglycemic clamp studies (Fig. 1d, e and Supplementary Fig. 2), p110α$^{DEL}$ mice displayed systemic insulin resistance at all of the time points tested (Fig. 2a and Supplementary Fig. 3). However, glucose tolerance was essentially unaffected throughout the life course. The lack of any effect seen in mice expressing only Adipoq–Cre, and lacking targeted p110α alleles, under the same experimental conditions, attests that these phenotypes can confidently be attributed to the deletion of p110α alleles (Supplementary Fig. 4). A number of hormones and cytokines and lipid profiles tested in the plasma of fasted one year old mice did not show any prominent differences, with the exception of substantially reduced insulin and leptin and slightly reduced adiponectin, and increased total cholesterol in the plasma of male p110α$^{DEL}$ mice (Fig. 2b–d). Notably, β-AR agonists have previously been reported to reduce production and secretion of adiponectin from the adipose tissue[16]. As both leptin and adiponectin are known to influence insulin sensitivity of metabolic tissues, reduced levels of leptin and adiponectin might play a role in the development of insulin resistance in liver and skeletal muscle, as seen in p110α$^{DEL}$ mice (Supplementary Fig. 2). Also, the differential impact of p110α inactivation on fasting insulin levels between male and female mice (Fig. 2b) suggests a sexually dimorphic phenotypic effect. Sex differences in the function of the adipose tissue are well-known in both mice and humans[17,18]. Such effects could account for a differential impact of p110α inactivation on insulin sensitivity and thus on fasting insulin levels between two sexes at the age point of this measurement. ITTs were also in line with female p110α$^{DEL}$ mice being substantially more insulin resistant than their control littermates, in

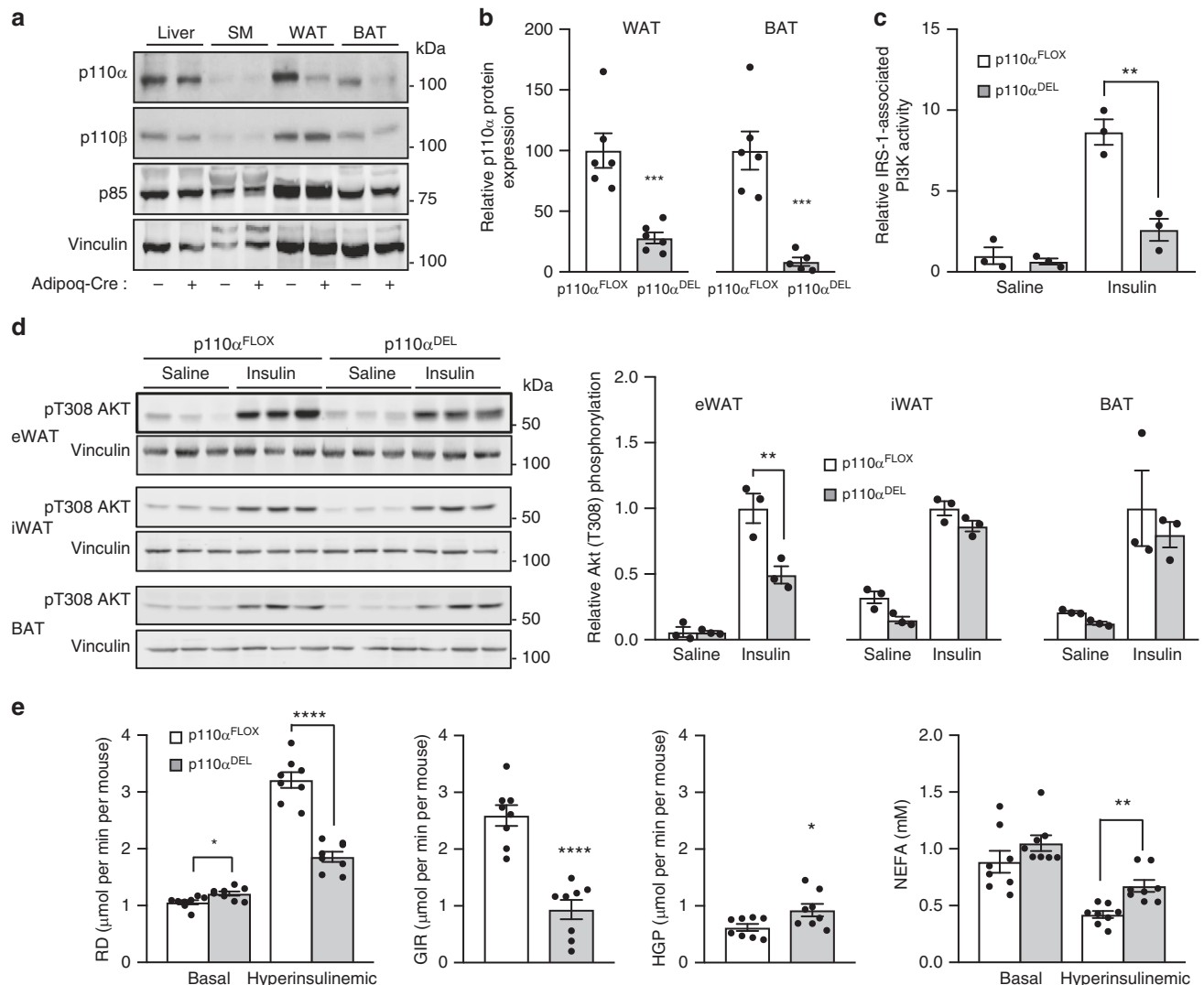

**Fig. 1** Adipose tissue-specific p110α^DEL mice are insulin resistant. **a** Representative immunoblot analysis of p110α protein levels in homogenates of metabolic tissues isolated from approximately 1.5-year-old male mice carrying or not the Adipoq–Cre transgene. Blots were also probed for the widely expressed p110β isoform and the PI3K regulatory subunit p85. Vinculin was probed as loading control. SM skeletal muscle (gastrocnemius), WAT white adipose tissue (epididymal), BAT brown adipose tissue (interscapular). **b** Semi-quantitative data for p110α expression in WAT and BAT ($n = 6$ mice per genotype). The respective blots are shown in Supplementary Fig. 1. **c**, **d** One-year-old mice ($n = 3$ per genotype) were injected intraperitoneally with 100 mU g$^{-1}$ of insulin. After 30 min, tissues were excised and snap frozen. **c** Adipose tissue (epididymal) lysates were immunoprecipitated with IRS-1 antibodies and assayed for associated PI3K lipid kinase activity. **d** Immunoblot analysis of adipose tissue (epididymal, eWAT; inguinal, iWAT; interscapular brown, BAT) lysates probed for Akt T308 phosphorylation. Each lane represents an individual mouse. **e** Hyperinsulinemic–euglycemic clamp analysis in 20-week-old female mice. Rate of glucose disappearance (RD), glucose infusion rate (GIR) and hepatic glucose production (HGP) at the hyperinsulinemic state and plasma levels of non-esterified fatty acids (NEFA) are shown. Average body weights: 21.7 and 21.6 g for p110a^DEL and p110a^FLOX mice ($n = 8$ per genotype), respectively. Data are presented as mean ± SEM. Statistical analysis: unpaired two-tailed $t$ test. $^*p < 0.05$; $^{**}p < 0.01$; $^{***}p < 0.001$; $^{****}p < 0.0001$

contrast to male mice, at this age point of 12 months (Supplementary Fig. 3).

When mice were subjected to high-fat feeding (45% calories derived from fat), weight gain was identical between the two genotypes (Fig. 3a). However, high-fat fed p110α^DEL mice were significantly more glucose tolerant despite being as insulin sensitive as p110α^FLOX littermates (Fig. 3b, c). To gain further insight into this unexpected effect of p110α inactivation, mice were subjected to a tissue-specific glucose uptake assay over a glucose tolerance test. We found that the BAT of p110α^DEL mice cleared approximately twice as much glucose from the circulation as p110α^FLOX littermates (Fig. 3d). No significant differences were detected in other organs, with the exception of visceral WAT, which in p110α^DEL mice cleared half as much glucose as

control littermates. High-fat feeding is known to activate BAT and increase expression of UCP1[19]. Consistent with higher BAT activation in p110α^DEL mice, we found that UCP1 protein expression was higher in p110α^DEL mice at basal level and remained higher upon high-fat feeding over 4 weeks (Fig. 3e, f and Supplementary Fig. 5). These data suggest that enhanced-BAT activity can counter the effects of insulin resistance on glucose homoeostasis of p110α^DEL mice.

**p110α inactivation potentiates β-adrenergic/cAMP signalling.** Since it is well-established that catecholamine stimulation of adrenergic signalling regulates glucose and lipid metabolism in BAT[20], the enhanced glucose clearance in the face of insulin

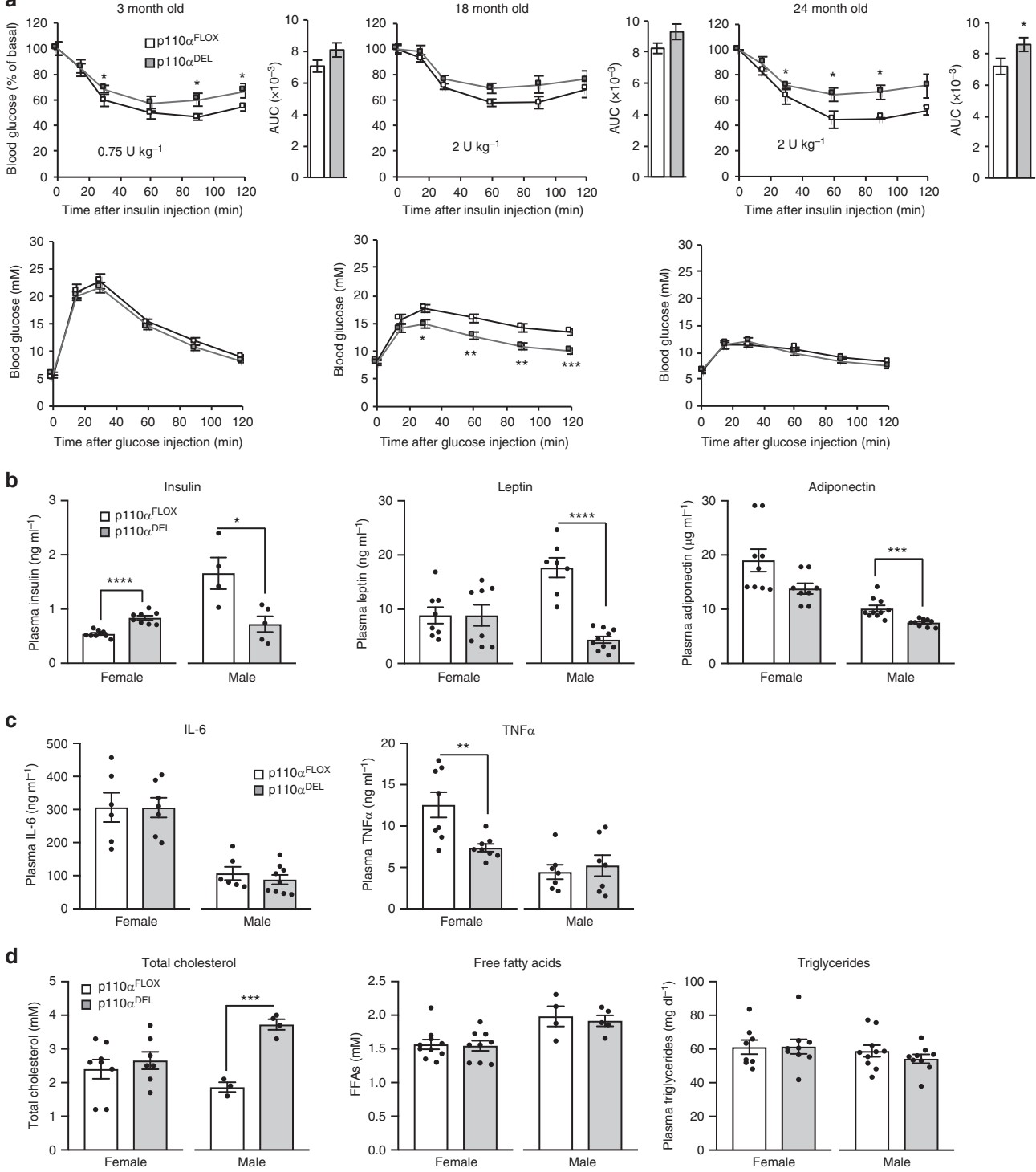

**Fig. 2** Insulin resistant p110α^DEL mice maintain normal glucose tolerance over ageing. **a** Cohorts of p110α^DEL and p110α^FLOX male mice were subjected to intraperitoneal glucose and insulin tolerance tests at various time points over ageing. For glucose tolerance test a bolus of glucose (2 g per kg of body weight) was injected intraperitoneally. Insulin doses were adjusted according to the age of the mice (0.75 U kg^−1 for 3-month-old and 2 U kg^−1 for 18- and 24-month-old). $n = 8$ per genotype at 3 months, 16 per genotype at 18 months, 14 per genotype at 24 months. AUC, area under curve. Additional age points and data for females are shown in Supplementary Fig. 3. **b** Fasting plasma levels of metabolic hormones in 1-year-old p110α^DEL and p110α^FLOX mice. n (p110α^FLOX/p110α^DEL), Insulin: Female, 9/8; Male, 4/5. Leptin: Female, 8/8; Male, 7/10. Adiponectin: Female, 8/8; Male, 10/10. **c** Fasting plasma levels of pro-inflammatory cytokines (IL6 and TNFa) in 1-year-old p110α^DEL and p110α^FLOX mice. n (p110α^FLOX/p110α^DEL), IL-6: Female, 6/7; Male, 6/9. TNFα: Female, 8/8; Male, 7/7. **d** Fasting plasma levels of lipids (total cholesterol, free fatty acids and triglycerides) in 1-year-old p110α^DEL and p110α^FLOX mice. n (p110α^FLOX/p110α^DEL), Cholesterol: Female, 8/7; Male, 3/4. FFAs: Female, 10/9; Male, 4/5. Triglycerides: Female, 8/9; Male, 10/9. n (p110α^FLOX/p110α^DEL), Cholesterol: Female, 8/7; Male, 3/4. FFAs: Female, 10/9; Male, 4/5. Triglycerides: Female, 8/9; Male, 10/9. Data are presented as mean ± SEM. Statistical analyses: Unpaired two-tailed *t* test. *p < 0.05; **p < 0.01; ***p < 0.001; ****p < 0.0001

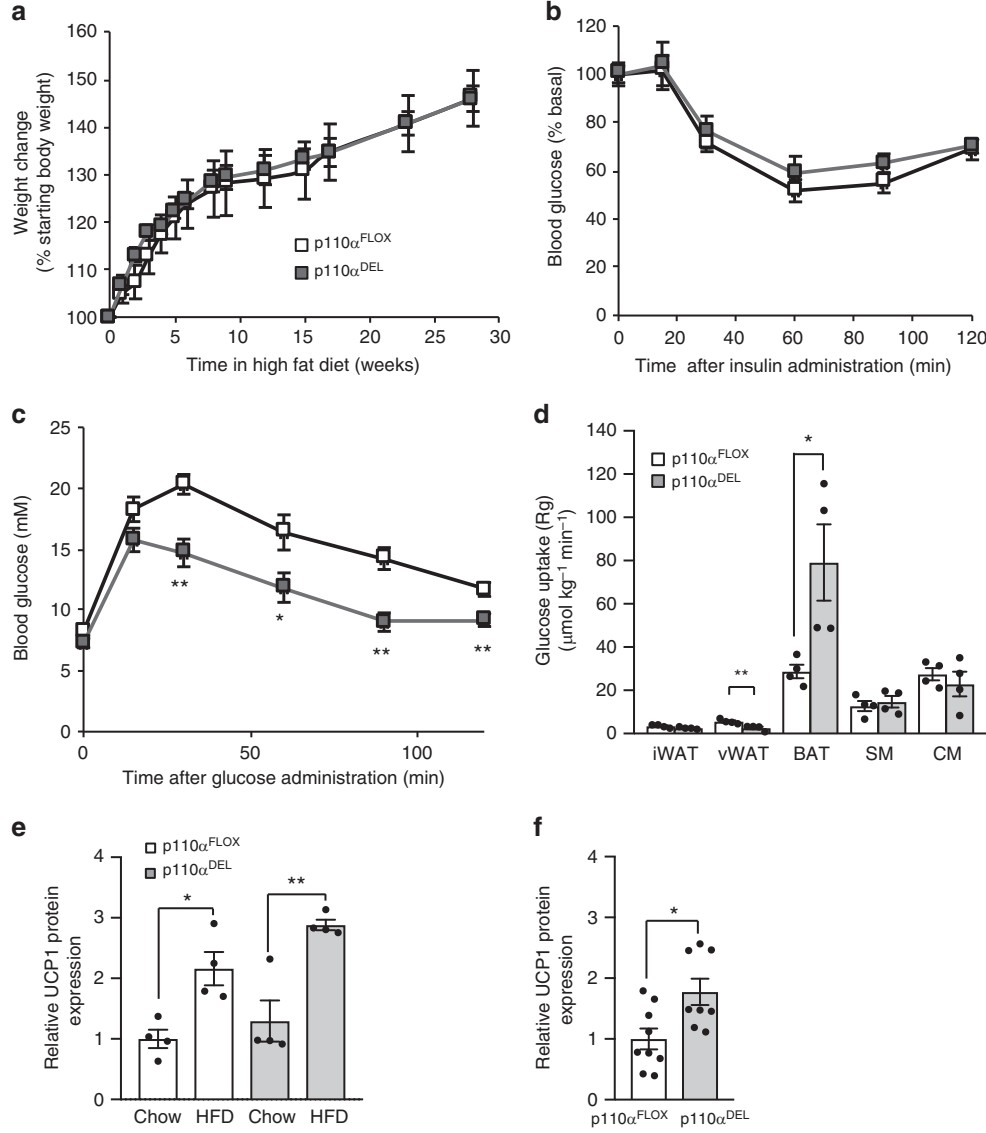

**Fig. 3** Increased glucose clearance from BAT of p110α$^{DEL}$ mice. **a** Weight gain curves of p110α$^{FLOX}$ and p110α$^{DEL}$ littermates. Thirty-week-old male mice ($n = 6$ per genotype) were fed a high-fat diet (HFD, 45% of calories from fat) for 28 weeks. **b, c** ITT (**b**) and GTT (**c**) in HFD-fed mice performed by intraperitoneal administration of insulin (2 U kg$^{-1}$) or glucose (1 g kg$^{-1}$). **d** HFD-fed mice were injected intraperitoneally with 1 g per kg glucose spiked with 2-[$^3$H]-deoxyglucose. Tissues were isolated after 1 h and tissue-specific glucose uptake was measured as described in Methods. SM skeletal muscle, CM cardiac muscle. **e** Expression of UCP1 protein in BAT of mice fed a high-fat diet (HFD, 60% of calories from fat) for 4 weeks ($n = 4$ per genotype and diet). Quantitative data from a blot shown in Supplementary Fig. 5a. **f** Basal levels of UCP1 protein expression in BAT of p110α$^{FLOX}$ and p110α$^{DEL}$ mice ($n = 9$ p110α$^{FLOX}$ and 8 p110α$^{DEL}$). Quantitative data from blots shown in Supplementary Fig. 5b. Data are presented as mean ± SEM. Statistical analysis: unpaired two-tailed $t$ test (**c, d, f**); one-way ANOVA with Tukey's multiple comparisons test (**e**). $^*p < 0.05$; $^{**}p < 0.01$

resistance prompted us to investigate the impact of p110α inactivation on the activity of adrenergic pathways in response to catecholamine stimulation. Furthermore, adrenergic pathways control fat metabolism by modulating lipolysis in adipose tissues[21]. β-Adrenergic receptors (β-ARs) activate adenylylcyclase through coupling to Gs subunits of heterotrimeric G proteins, resulting in the elevation of intracellular levels of cAMP and activation of its effector molecules, notably of protein kinase A (PKA).

Explants from subcutaneous (inguinal) and visceral adipose tissue were stimulated with the synthetic β-AR agonist isoproterenol or the natural catecholamine norepinephrine. As readouts of β-AR/cAMP pathway activation, we assessed phosphorylation of hormone sensitive lipase (HSL), perilipin and cAMP response element binding (CREB) protein, all of which are targets of PKA

in the adrenergic pathway in adipocytes. Tissue explants isolated from subcutaneous adipose tissue of p110α$^{DEL}$ mice, as well as those isolated from p110α$^{FLOX}$ mice and treated with the p110α selective inhibitor A66[22], displayed enhanced sensitivity to β-AR stimulation by both isoproterenol (Fig. 4a, b) and norepinephrine (Supplementary Fig. 6a, b). However, a similar effect was not the case in explants isolated from visceral adipose tissue. Consistent with the observed increase in the phosphorylation of lipolysis regulatory molecules upon p110α inactivation, release of glycerol, which provides an index of the rate of lipolysis (Fig. 4c and Supplementary Fig. 6c), and intracellular cAMP levels (Fig. 4d and Supplementary Fig. 6d) were found to be increased in the subcutaneous adipose tissue of both p110α$^{DEL}$ mice and that of p110α$^{FLOX}$ mice treated with A66.

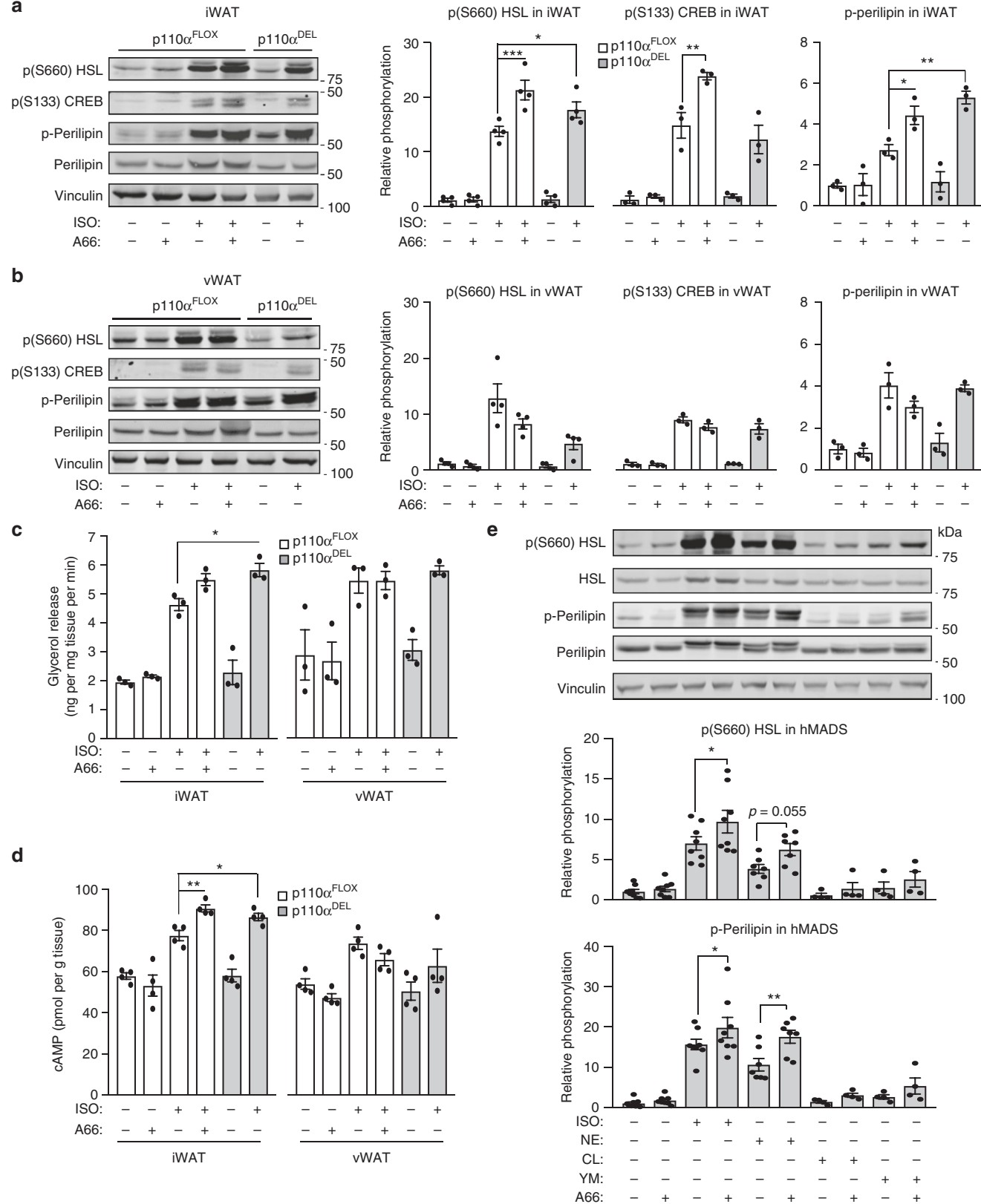

Since differences in the expression of various adrenergic receptor classes and subtypes have been reported between mouse and human adipose tissue[23], we tested whether inhibition of p110α has a similar potentiating effect on adipocytes differentiated from human multipotent adipose-derived stem (hMADS) cells[24]. As with mouse adipocytes, treatment of hMADS cell-derived adipocytes with A66 resulted in potentiation of signalling induced by a number of β-AR agonists (Fig. 4e).

**Mechanism of adrenergic signalling enhancement.** We sought to gain further insights in the molecular mechanism underlying

**Fig. 4** Inactivation of p110α potentiates β-adrenergic signalling selectively in iWAT. **a, b** Explants of inguinal (iWAT) (**a**) and visceral (vWAT) (**b**) white adipose tissue were isolated from 1-year-old p110α^DEL and p110α^FLOX littermates and stimulated ex vivo with 1 μM isoproterenol (ISO) in the presence or absence of the p110α inhibitor A66 (2 μM) for 20 min, followed by tissue homogenisation and immunoblot analysis of phosphorylation of HSL, perilipin and CREB. Phosphorylation of perilipin was assessed using phospho-PKA substrate antibodies. Representative immunoblots are shown. Graphs show quantitative data pooled from $n = 4$ (pHSL) and $n = 3$ (pCREB, p-Perilipin) independent experiments. **c** Lipolysis in explants from 1-year-old male mice ($n = 3$ per genotype) assessed as glycerol release over 30 min following stimulation with 1 μM ISO in the presence or absence of 2 μM A66. **d** cAMP levels measured by ELISA in explants from approximately 1-year-old male mice ($n = 4$ per genotype) stimulated with 1 μM ISO in the presence or absence of 2 μM A66 for 20 min. **e** Inhibition of p110α potentiates adrenergic signalling in hMADS cell-derived adipocytes stimulated with non-selective (ISO isoproterenol, NE norepinephrine) or β3-selective (CL, CL316,243; YM, YM178) β-AR agonists at 1 μM. Quantitative data from $n = 8$ for basal−/ A66−/ISO−/ISO+ A66-treated, $n = 7$ for NE−/NE+ A66-treated, $n = 4$ for CL−/CL+ A66− and YM−/YM+ A66-treated independent experiments and a representative immunoblot are shown in (**e**). Data are presented as mean ± SEM. Statistical analyses: one-way ANOVA with Bonferroni's multiple comparisons test (**a–d**) and paired two-tailed $t$ test (**e**). $^{*}p < 0.05$; $^{**}p < 0.01$; $^{***}p < 0.001$

the potentiation of adrenergic signalling by p110α inhibition. To this end, we probed the levels of β-AR subtype expression in adipose tissues of p110α^FLOX and p110α^DEL mice. We found no significant differences in the expression of the various β-AR subtypes among the different depots tested (Supplementary Fig. 7). At the level of PI3K isoform expression, we observed similar levels of expression of p110α between iWAT and vWAT, but a tendency for higher level of p110β in iWAT compared to vWAT (Supplementary Fig. 7). This is consistent with the milder effect of p110α deletion on insulin sensitivity in iWAT compared to vWAT (Fig. 1), as higher levels of p110β could partially compensate for p110α deficiency in iWAT.

Insulin exerts its anti-lipolytic action mainly, though not exclusively, through PI3K/Akt-mediated phosphorylation and activation of the cAMP degrading enzyme phosphodiesterase 3B (PDE3B)[25,26]. We tested whether β-adrenergic stimulation can also activate Akt phosphorylation in iWAT and vWAT. We found that isoproterenol stimulation induced a substantial increase in Akt phosphorylation in both iWAT and vWAT explants (Fig. 5a). We also tested immortalised brown adipocytes and BAT explants derived from p110α^FLOX mice. Similar to iWAT, inhibition of p110α in brown adipocytes (Fig. 5b) and in BAT explants (Fig. 5c) potentiated NE-stimulated (immortalised brown adipocytes) or ISO-stimulated (BAT explants) adrenergic signalling, as assessed by phosphorylation of HSL and of perilipin. p110α inhibition also potentiated phosphorylation of p38 MAPK, which has previously been shown to be an important mediator of cAMP-dependent induction of UCP1 expression in brown adipocytes[27]. Similar to WAT, NE-stimulated Akt phosphorylation in brown adipocytes in a PI3K dependent manner (Fig. 5d). Notably, the effect of p110α inhibitor A66 on Akt phosphorylation in brown adipocytes was substantially lower than that in WAT and a much stronger effect was achieved upon combined treatment with A66 and a p110β-selective inhibitor (TGX221). This finding suggests that in addition to p110α, the other broadly expressed PI3K isoform, p110β, plays a role in NE-stimulated PI3K signalling in BAT. Consistent with reduced Akt activation, inhibition of PI3K reduced NE-stimulated phosphodiesterase activity in extracts of p110α^FLOX or p110α^DEL adipocytes (Fig. 5e). In this regard, our data are consistent with the previously reported catecholamine-induced phosphorylation of Akt and PDE3B activation in adipocytes[28]. Since long-term insulin and catecholamine treatment has previously been shown to alter PDE3B expression levels[29], we also tested whether reduced expression of PDE3B might account for enhanced adrenergic signalling in p110α^DEL adipocytes. However, protein levels of PDE3B were only marginally lower in extracts of p110α^DEL iWAT (Supplementary Fig. 8). Furthermore, in line with previous reports[30], we found that activation of PI3K in the context of β-AR signalling is likely mediated through the key cAMP effector EPAC, as treatment of

brown adipocytes with the EPAC inhibitor ESI09 tended to reduce NE-stimulated Akt phosphorylation (Fig. 5d) and prevented NE stimulation of phosphodiesterase activity (Fig. 5e). In summary, adrenergic stimulation activates two opposing activities in the adipose tissue: adenylylcyclase that produces cAMP and stimulates the PKA pathway and phosphodiesterase that degrades cAMP. Therefore, inhibition of p110α can be potentiating adrenergic signalling by limiting the concomitant activation of PDE activity upon adrenergic stimulation (Fig. 5f). Consistent with this, inhibition of p110α with A66 fails to increase adrenergic signalling above the level attained through treatment with the general phosphodiesterase inhibitor isobutyl-methylxanthine (Supplementary Fig. 9).

**Increased NE-stimulated EE in p110α^DEL mice.** Adrenergic stimulation of adipose tissue results in the recruitment of 'brown-like' (*beige* or *brite*) adipocytes in WAT[31]. In order to further investigate the effect of p110α inhibition in the adipose tissue in vivo, we administered CL316,243, a β3-AR-specific agonist, to 18-week-old mice daily for a period of 5 days. As shown in Fig. 6a, administration of CL316,243 increased the mRNA expression of *Ucp1* and other thermogenic genes, markers of brown and *beige* adipocytes, specifically in the inguinal WAT of p110α^FLOX and p110α^DEL mice. Increased expression of *Ucp1* mRNA was accompanied by increased protein levels as determined by immunoblot analysis (Fig. 6b, c and Supplementary Fig. 10) and immunohistochemistry (Fig. 6d). In line with the above documented increase in adrenergic signalling in the subcutaneous WAT of p110α^DEL mice, a higher expression of all these markers was observed in the inguinal WAT of p110α^DEL compared to p110α^FLOX mice. This suggests that inactivation of p110α potentiates adrenergic signalling in vivo and increases recruitment of *beige* adipocytes upon adrenergic stimulation of subcutaneous adipose tissue.

To complete the physiological phenotypic analysis of p110α^DEL mice, we performed indirect calorimetry to determine EE under free-living conditions, as well as in response to catecholamine stimulation. As shown in Fig. 6e, f, EE and respiratory exchange ratio were similar between approximately 1-year-old p110α^FLOX and p110α^DEL mice under free-living conditions. These experiments were performed in weight-matched groups of mice to avoid the confounding effects of different metabolic mass. No differences in water and food consumption or in loss of body weight over the time course of the experiment were observed between the genotypes (Fig. 6g). However, injection of norepinephrine into unconscious mice measured at thermoneutrality (30 °C) showed a significantly increased EE in p110α^DEL mice compared to p110α^FLOX littermates (Fig. 6h, i). p110α^DEL mice also tended to display a slightly higher basal EE, which was consistent with their higher basal glucose turnover during the

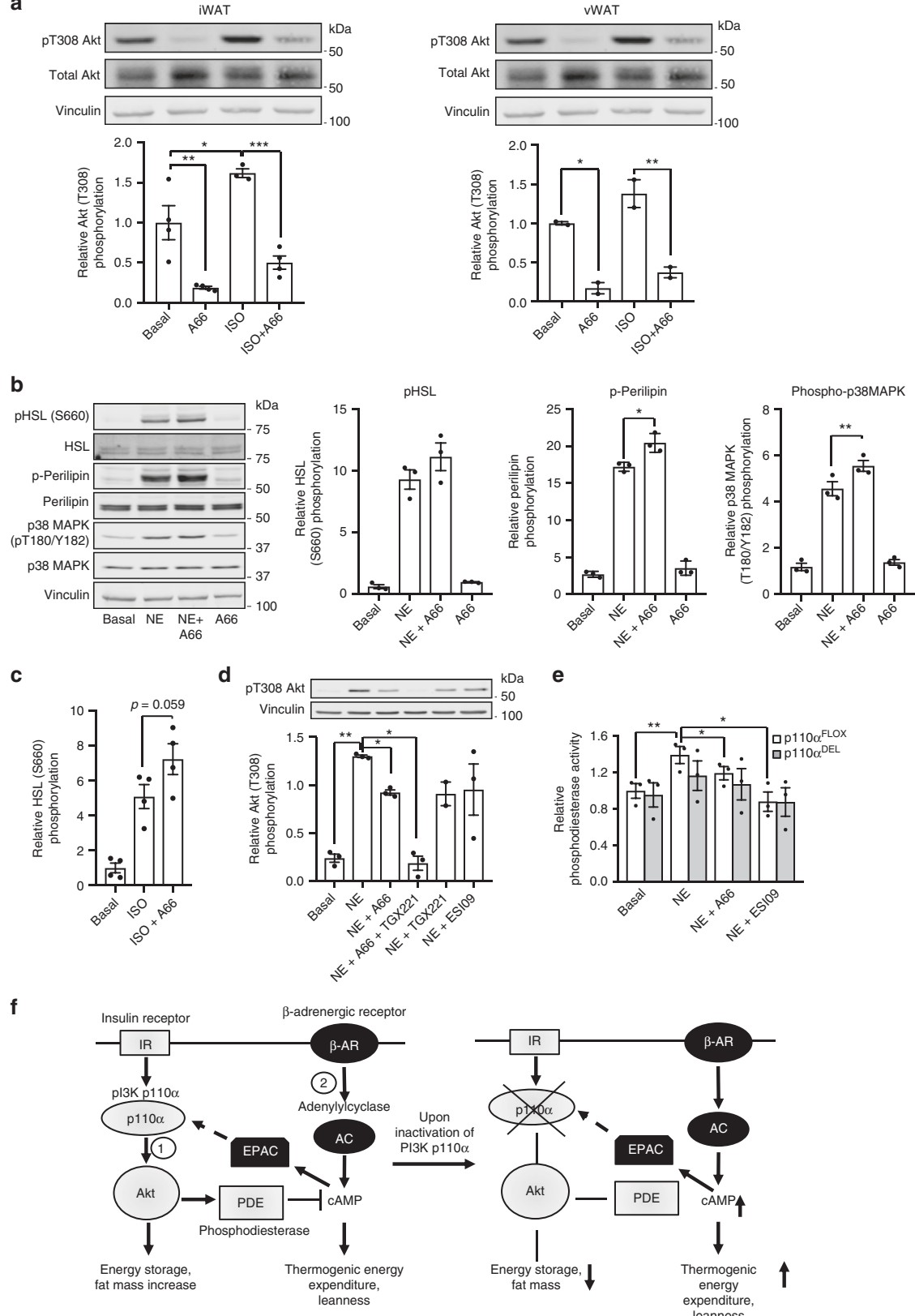

clamp studies (Fig. 1f), but this did not reach statistical significance. The weights of all the mice were very similar (Fig. 6j), whereas basal EE was 10% larger in p110α$^{DEL}$ mice, suggesting that changes in EE were not simply a product of

changes in body weight and therefore metabolic mass. These findings are consistent with the enhanced β-adrenergic signalling in isolated adipose tissue explants from p110α$^{DEL}$ mice (Fig. 4 and Supplementary Fig. 6).

**Fig. 5** Mechanisms of adrenergic signalling enhancement by p110α inhibition. **a** β-AR stimulation induces p110α dependent Akt activation. Immunoblot analysis of Akt phosphorylation induced by treatment with 1 µM isoproterenol in the presence or absence of 2 µM A66 in iWAT and vWAT explants isolated from 38-week-old male p110α$^{FLOX}$ mice. Graphs show quantitative data from iWAT of three mice and vWAT of two mice. A representative immunoblot is shown. **b–e** Inhibition of p110α potentiates β-adrenergic signalling in immortalised BAT-derived adipocytes (**b**) and BAT explants (**c**). Immunoblot detection of NE-stimulated (1 µM for 20 min) HSL (S660), perilipin and p38MAPK (T180/Y182) phosphorylation in lysates of BAT-derived adipocytes (**b**) or isoproterenol (ISO)-stimulated (1 µM for 20 min) BAT explants (**c**). Representative immunoblots and graphs showing average data from three ($n = 3$) cell pools or explants isolated from individual p110α$^{FLOX}$ mice are shown. **d** NE-stimulated (1 µM for 10 min) Akt (T308) phosphorylation in the presence of inhibitors against PI3K p110α (A66, 2 µM) and p110β (TGX221, 1 µM) and EPAC (ESI09, 10 µM). A representative blot and data from three ($n = 3$) BAT-derived cell pools isolated from different p110α$^{FLOX}$ mice are shown. **e** Phosphodiesterase activity measured in lysates of NE-stimulated (1 µM for 20 min) BAT-derived cell pools ($n = 3$ per genotype) isolated from individual p110α$^{FLOX}$ and p110α$^{DEL}$ mice in the presence or absence of p110α inhibitor (A66, 2 µM) or EPAC inhibitor (ESI09, 10 µM). **f** Role of PI3K p110α in insulin- and β-adrenergic receptor-stimulated pathways in adipocytes. Upon activation of the Insulin Receptor (IR), PI3K p110α promotes activation of phosphodiesterase (PDE) through phosphorylation by Akt (1). p110α-dependent and Akt-mediated activation of PDE also occurs upon β-AR stimulation, possibly through EPAC activation, concomitantly with activation of adenylylcyclase (AC) (2). According to scheme, inhibition of p110α reduces energy storage and fat mass increase through the insulin pathway and promotes thermogenic energy expenditure and leanness through the β-AR pathway. Data are presented as mean ± SEM. Statistical analysis: one-way ANOVA with Tukey's multiple comparisons test (**a**); paired two-tailed $t$ test (**b**, **c**); one-way ANOVA with Bonferroni's multiple comparisons test (**d**, **e**). $^*p < 0.05$; $^{**}p < 0.01$; $^{***}p < 0.001$

**Reduced age-dependent fat accumulation in p110α$^{DEL}$ mice**. We generated cohorts of p110α$^{FLOX}$ and p110α$^{DEL}$ littermates and monitored their weight gain over time. As shown in Fig. 7a, b, p110α$^{DEL}$ mice were born with a body weight similar to that of p110α$^{FLOX}$ littermates. However, their growth curves started to diverge at around 8 months of age and p110α$^{DEL}$ mice did not accumulate as much weight as p110α$^{FLOX}$ littermates with age. More specifically, the average weight of p110α$^{DEL}$ mice for the period between 8–26 months of age was 7 and 8% lower for male and female mice, respectively, compared to their p110α$^{FLOX}$ littermates. Gonadal WAT and interscapular BAT weights were also reduced in p110α$^{DEL}$ mice at 1.5 years of age, whereas BAT masses normalised to body weight were similar in 2-year-old mice, indicating a slower degeneration rate in BAT of p110α$^{DEL}$ mice (Fig. 7c). Assessment of food intake in 1-year-old mice showed no significant differences in food intake between p110α$^{DEL}$ and p110α$^{FLOX}$ littermates (Fig. 7d, e), except for the tendency of p110α$^{DEL}$ male mice to consume slightly more food (Fig. 7e), demonstrating that the weight difference between the two genotypes was not due to different levels of food consumption. Therefore, the increased catecholamine-stimulated EE documented in our physiological studies described above could be reducing fat accumulation over the life course in p110α$^{DEL}$ mice.

## Discussion

Metabolism is an essential process for all organisms and, like many other biological processes, is adversely affected by ageing. However, this relationship is also valid in the inverse direction in that metabolic deterioration accelerates the rate of ageing in a vicious cycle. In fact, glucose and lipid homoeostasis are emerging as key determinants of the rate of ageing in a diverse range of animal organisms. Consistent with this, genetic and pharmacological interventions affecting signalling pathways that regulate metabolism, notably the IIS, mTOR and AMPK pathways, have been shown to affect the lifespan of various organisms[32]. The fact that downregulation of essential signalling pathways, such as IIS and mTOR, exerts beneficial effects is seemingly paradoxical. In the context of metabolism, the gross phenotypic effects of inactivation of these pathways are in many cases initially adverse, consistent with the importance of these pathways, and beneficial effects emerge only later in life. It is therefore essential that longitudinal studies are conducted in model organisms in order to assess the long-term effects of such interventions and to decipher the molecular mechanisms underlying these beneficial

effects. These insights will help to reconcile the paradox of beneficial effects ensuing from the downregulation of essential metabolic pathways.

Here, we report the findings of a longitudinal metabolic study in mice that characterised the effects of adipose tissue-specific deletion of PI3K p110α, the partial global inactivation of which has been shown to confer a beneficial effect later in life and to moderately increase lifespan in mice[12]. We found that adipose tissue-specific inactivation of p110α potentiated adrenergic signalling, specifically in subcutaneous white adipose tissue and in brown adipose tissue. The molecular mechanism underlying this effect appears to involve engagement of p110α in catecholamine-stimulated Akt phosphorylation, which in turns activates phosphodiesterase and degradation of cAMP. Deletion of p110α reduced catecholamine-induced PDE activation thus enhancing cAMP-mediated signalling. The significance of cAMP-mediated signalling in thermogenic conversion of white adipose tissue has recently been demonstrated by knockout of PDE3B shown to promote browning of epididymal white adipose tissue[33].

The potentiating effect on adrenergic signalling observed upon p110α inactivation was modest and most likely for this reason insufficient to confer protection from weight gain during high-fat feeding. However, glucose uptake by BAT was higher in p110α$^{DEL}$ mice fed a high-fat diet, despite their similar insulin sensitivity with p110α$^{FLOX}$ mice. This increased glucose uptake from BAT likely explains the normal glucose tolerance in the face of systemic insulin resistance documented over ageing of p110α$^{DEL}$ mice. Moreover, p110α$^{DEL}$ mice did not accumulate as much fat as p110α$^{FLOX}$ littermates upon ageing, which is arguably a more physiological type of metabolic stress than high-fat feeding, demonstrating that the modest effect of p110α deletion on adrenergic signalling is sufficient to confer a beneficial effect under physiological conditions. Indeed, there is substantial evidence demonstrating that ageing diminishes the sensitivity of adrenergic stimulation to reduce EE in humans[34,35] as well as in rodents[36]. In this regard, p110α inactivation could be counteracting the negative impact of ageing via regulating adrenergic signalling in the adipose tissue and in particular in BAT where it increased levels of UCP1.

Recently, another study reported a phenotype of adipose tissue-specific p110α deletion using the aP2–Cre line[37]. The reported phenotype is fundamentally different to the phenotype described here and includes a decrease in thermogenic capacity of the adipose tissue, as a result of downregulation of UCP1 and consequent development of obesity and liver steatosis. The reason for this striking discrepancy could be related to the different

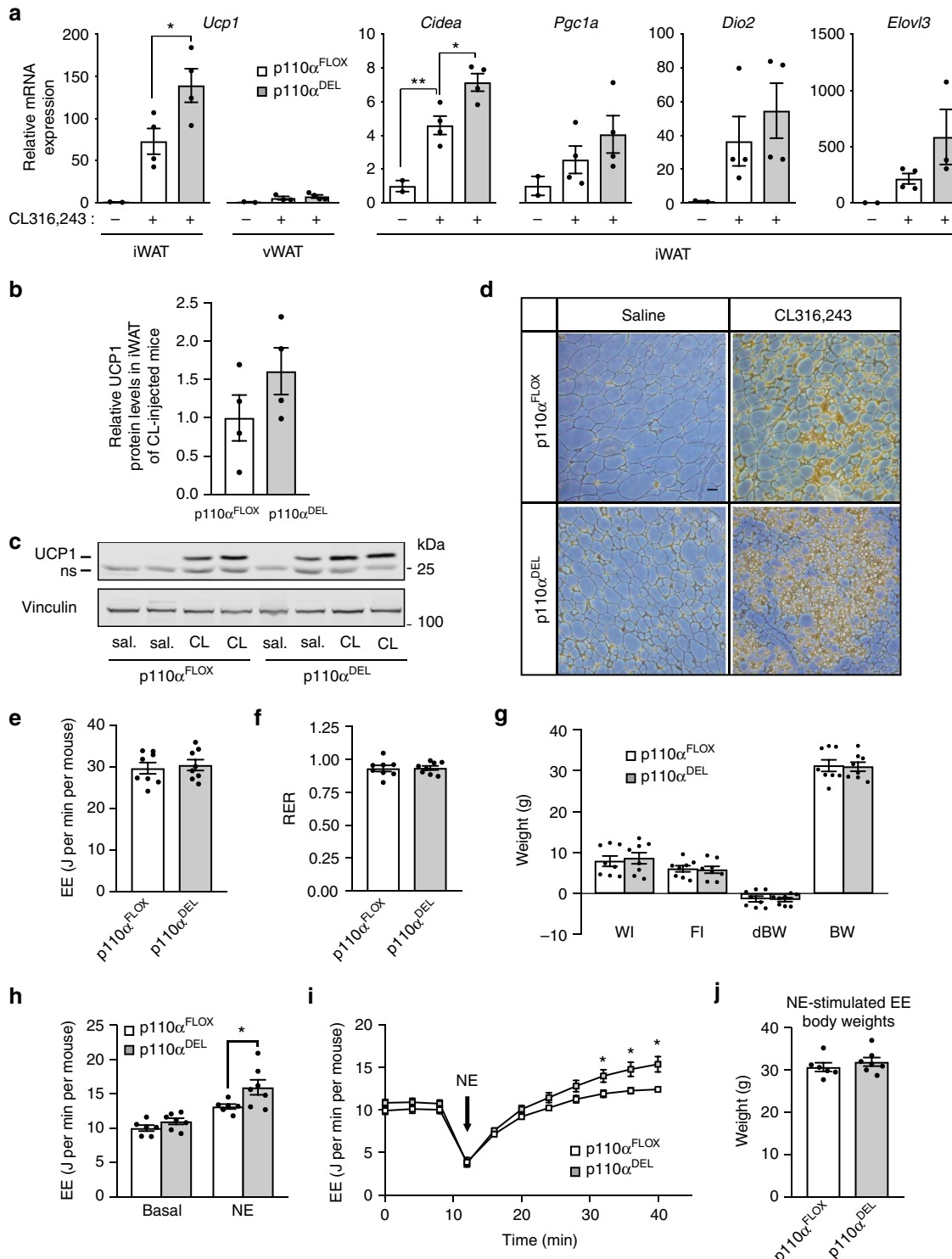

conditional p110α alleles or the different Cre lines (aP2–Cre versus adiponectin–Cre) used in these two studies. Also, potential differences in the level of p110α deletion could critically determine the nature of the ensuing phenotypes. In the present study, we documented a reduction in p110α protein levels of approximately 70–90%, depending on the tissue. However, it is likely that higher levels of p110α deletion could result in more severe insulin resistance, of which the negative impact would outweigh the potential benefits. Nevertheless, the phenotype described in the present study is consistent with the beneficial metabolic effects of deletion of the IR in the adipose tissue of FIRKO mice[9] as well

as with the recent demonstration that administration of a dual p110α/δ PI3K inhibitor protects mice and monkeys from diet-induced obesity and metabolic syndrome[38], an effect more recently shown to be largely recapitulated by administration of a p110α selective inhibitor[39]. In the FIRKO mice, an increase in oxidative metabolism in the adipose tissue was reported[40], and in the latter model, PI3K inhibition was shown to increase the levels of UCP1 expression and fuel oxidation in BAT[41], in line with our findings.

The physiological importance of adrenergic signalling in the development of age-related obesity has recently been highlighted

**Fig. 6** p110α$^{DEL}$ mice display increased catecholamine-stimulated energy expenditure. **a** mRNA-expression levels of brown adipocyte markers in adipose tissues isolated from 18-week-old p110α$^{DEL}$ and p110α$^{FLOX}$ littermate male mice ($n = 4$ per genotype) following intraperitoneal administration of the β$_3$-AR selective agonist CL316,243 (1 mg per kg per day) over five consecutive days. RNA was extracted from inguinal and epididymal adipose tissues and mRNA expression of *Ucp1*, *Cidea*, *Pgc1a*, *Deiodinase 2* (*Dio2*) and *Elovl3* was analysed by quantitative PCR. **b** UCP1 protein expression detected by immunoblot analysis (blot shown in Supplementary Fig. 10) in iWAT from the same mice as in (**a**) ($n = 4$ per genotype). **c, d** Immunoblot analysis (**c**) and immunohistochemical detection (**d**) of UCP1 protein expression in iWAT of 6-week-old male mice injected with CL316,243 as in (**a**) (ns non-specific band). Scale bar: 50 μm. **e** Raw energy expenditure of approximately 1-year-old p110α$^{FLOX}$ and p110α$^{DEL}$ female mice measured for 48 h at room temperature. **f** Respiratory exchange ratio (RER) for the 48 h calorimetry run. **g** Water intake (WI), food intake (FI), change in body weight (dBW) and average body weight (BW) for the 48 h of the calorimetry run. **h–j** Norepinephrine (NE)-stimulated energy expenditure (EE). **h** Energy expenditure measured in unconscious mice housed at 30 °C prior to (basal) and after injection with 0.5 mg kg$^{-1}$ NE. Bars are averages of three stable readings for baseline prior to injection. NE-stimulated is the average of the three largest readings recorded. **i** Graph showing the effects of NE injection on energy expenditure with time. **j** Body weights of the mice used at the time they were analysed. **e–g** $n = 8$ mice per genotype; **h–j**, $n = 6$ p110α$^{FLOX}$ and 7 p110α$^{DEL}$ mice. Data are presented as mean ± SEM. Statistical analysis: one-way ANOVA with Bonferroni's multiple comparisons test (**a**); unpaired two-tailed *t* test (**h, i**). $^*p < 0.05$; $^{**}p < 0.01$

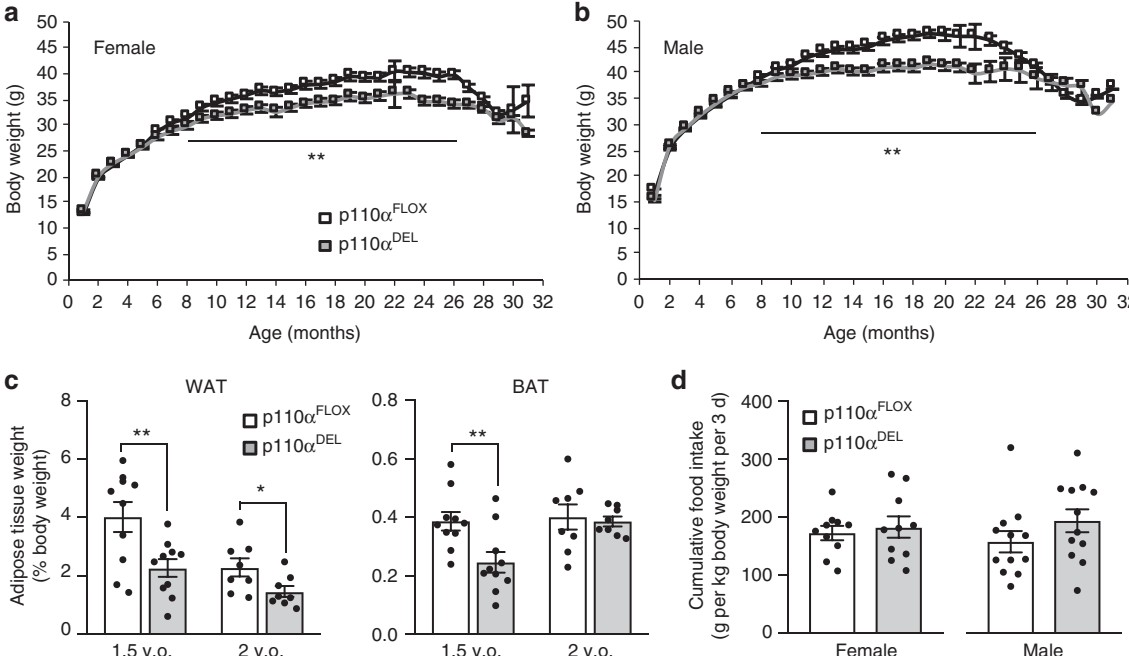

**Fig. 7** p110α$^{DEL}$ mice accumulate less weight with age. **a, b** Body weight gain over time in female (**a**) and male (**b**) p110α$^{DEL}$ and p110α$^{FLOX}$ mice. The average body weight of each group of mice between 8 and 26 months of age were compared by unpaired *t* test. $n = 66$ per genotype for male; $n = 76$ p110α$^{FLOX}$ and 58 p110α$^{DEL}$ for female. **c** Epididymal WAT and interscapular BAT mass of 1.5- and 2-year-old male mice. $n = 10$ per genotype for WAT; $n = 8$ per genotype for BAT. **d** Food intake of approximately 1-year-old mice. Each data point represents an average measurement from a single cage. Female mice (29 p110α$^{FLOX}$ and 20 p110α$^{DEL}$) were distributed in 10 cages per genotype ($n = 10$) and male mice (22 p110α$^{FLOX}$ and 24 p110α$^{DEL}$) mice were distributed in 12 cages per genotype ($n = 12$). Data are presented as mean ± SEM. Statistical analysis: unpaired two-tailed *t* test. $^*p < 0.05$; $^{**}p < 0.01$

by the demonstration that inflammatory macrophages reduce adrenergic output by degrading catecholamines in the adipose tissue[42,43]. An important implication of the present study is that potentiation of adrenergic signalling in adipose tissue could be a promising therapeutic approach to promote adipose tissue browning and increased thermogenic EE, processes which have been pursued in the therapy of obesity[44,45]. The use of β$_3$-AR agonists as potential therapeutics for obesity and type-2 diabetes has been pursued intensively in the recent past, but lack of efficacy or side effects (mainly due to adrenergic stimulation of the cardiovascular system) have prevented translation to clinical application[46]. It is tempting to speculate that concomitant administration of a p110α inhibitor could potentiate the effect of β$_3$-adrenergic agonists, increasing their efficacy and their safety profile by reducing the effective doses, thus avoiding the side effects of cross-activation of other types of β-adrenergic receptors in other organs. YM178 (Mirabegron), a human β$_3$-AR selective agonist, which is clinically approved for other indications, has

recently shown efficacy in activation of human BAT[47]. Also, as mentioned above, administration of a PI3K inhibitor reversed diet-induced obesity in mice and monkeys, providing proof-of-principle of the utility of PI3K inhibitors in the context of metabolic disease treatment[38]. Moreover, a number of p110α inhibitors are undergoing clinical trials in oncology. Taken together, these suggest that testing combinations of p110α inhibitors and β$_3$-adrenergic agonists in the treatment of obesity is a realistic possibility.

Finally, the present work provides an example of how an intervention that negatively affects an essential signalling pathway, such as the IIS pathway, can at the same time act beneficially to mitigate the adverse effects, thus reconciling the insulin paradox at the mechanistic level.

## Methods

**Mouse strains and maintenance.** To generate mouse lines with adipose tissue-specific deletion of PI3K p110α, mice with conditionally targeted p110α alleles

(p110α[FLOX])[13] were crossed with transgenic mice expressing a Cre recombinase under the control of the adiponectin promoter (Adipoq–Cre)[14]. Both lines were in the C57BL6/J background. Experimental mice were littermates produced from crosses of Adipoq–Cre (+) p110α[FLOX] mice with Adipoq–Cre (−) p110α[FLOX] mice. Mice were housed in individually ventilated cages under a 12 h light/12 h dark cycle at constant temperature (20–23 °C) with ad libitum access to food and water and were fed a standard diet (2018, 18% Protein Rodent diet, Harlan Teklad) or a high-fat diet (45% calories from fat; 58V8, TestDiet).

**Maintenance and differentiation of hMADS cells.** hMADS-3 cells isolated form the prepubic fat pad of a 4-month-old male[48] were maintained in DMEM supplemented with 10% foetal bovine serum and 2.5 ng ml$^{-1}$ human fibroblast growth factors. Adipocyte diffentiation was induced 24 h postconfluence by replacing maintenance medium with differentiation induction medium consisting of DMEM/Ham's F12 (1:1) supplemented with 5 μg ml$^{-1}$ human insulin, 10 μg/ml$^{-1}$ human transferrin, 0.2 nM 3,3′,5-Triiodo-l-thyronine (T3), 1 μM rosiglitazone, 100 μM iso-butylmethyxanthine (IBMX) and 1 μM dexamethasone. Three days later, the medium was replaced with differentiation induction medium without IBMX and dexamethasone. Adipocytes were used in experiments upon completion of differentiation, typically 5–7 days after induction.

**Isolation, immortalisation and differentiation of brown pre-adipocytes.** Interscapular BAT was isolated from three p110α[FLOX] and three p110α[DEL] 5-week-old male mice, minced with fine scissors and digested with 1 mg ml$^{-1}$ collagenase for 1 h at 37 °C. Mature adipocytes were separated from the stromal vascular fraction (SVF) by centrifugation at 250$g$ for 5 min. Pellets containing the SVF were seeded in 6-well cell culture dishes in DMEM supplemented with 10% foetal bovine serum and transduced with retroviruses carrying a plasmid encoding SV40 Large T antigen (pBabe-puro-SV40LT, Addgene plasmid no 14088). Transduced cells were selected with puromycin (2 μg ml$^{-1}$) for 2 weeks. Adipocyte differentiation was induced by treating confluent cell monolayers with 1 μg ml$^{-1}$ insulin, 1 μM dexamethasone, 0.5 mM isobutylmethyxanthine (IBMX), 1 μM rosiglitazone and 1 nM 3,3′,5-Triiodo-l-thyronine (T3). After 48 h, the medium was replaced with fresh DMEM supplemented with 10% foetal bovine serum. Cell monolayers were used in experiments upon completion of differentiation, typically 1 week after induction.

**Body weight and food intake analysis.** Body weight was measured monthly from 4 weeks of age onwards. For food intake assays, mice were housed in small groups of 2–3 individuals and allowed to acclimatise for 2 days before the study. Food pellet and mouse body weights were measured at the beginning and end of a 3-day measurement. Results are expressed as cumulative food intake (grams of chow per kg of body weight per 3 days).

**Blood metabolic biomarkers.** Blood samples from fed or fasted (16 h) mice were collected from the tail vein with heparinized capillaries and then spun in a benchtop microcentrifuge for 5 min to pellet blood cells. Plasma was transferred to a fresh tube and stored at −80 °C until further use. Mouse plasma hormone and cytokine levels were determined by ELISA kits: Insulin, leptin and adiponectin (Crystal Chem); TNFα and Interleukin-6 (PeproTech). Triglycerides, free fatty acids and cholesterol in plasma were determined using the Infinity Triglycerides Liquid stable Reagent (TR22421; Thermo Scientific), NEFA-HR(2) kit and Cholesterol Liquid (Wako, Richmond, VA), respectively.

**Glucose and insulin tolerance tests.** GTTs and ITTs were performed on mice at different ages, fasted for 16 or 5 h, respectively. Blood glucose levels were determined at several time points (up to 120 min) after intraperitoneal injection of a glucose solution (2 g per kg of body weight) or human insulin (Actrapid, Novo Nordisk) (0.75–2 U per kg of body weight). Blood glucose concentrations were measured in tail vein blood samples using a Contour XT glucometer (Bayer).

**Hyperinsulinemic–euglycemic clamps.** Hyperinsulinemic–euglycemic clamps were performed in anaesthetised animals on thermally controlled pads to maintain their core body temperature. Animals were anesthetized by intraperitoneal injection of a combination of 6.25 mg per kg acetylpromazine, 6.25 mg kg$^{-1}$ midazolam and 0.31 mg kg$^{-1}$ fentanyl. An infusion needle was placed into the tail vein and 6,6-D2 D-glucose (Goss Scientific) was infused at a rate of 100 nM min$^{-1}$ for 80 min to achieve steady-state levels and blood samples collected on blood spot cards (Whatman 903 Protein Saver Card, SLS) to determine basal glucose turnover. Thereafter, insulin (Actrapid; Novo Nordisk) was infused at a constant rate of 0.91 mU min$^{-1}$ after a bolus dose of 16.5 mU and 6,6-D2 D-glucose infusion rate was increased to 200 nM min$^{-1}$ to limit changes in enrichment levels of labelled glucose in blood. A variable infusion of 12.5% D-glucose was used to maintain blood glucose at euglycemic (basal) levels. Blood glucose was measured with an AlphaTRAK2 glucometer (Abbott Animal Health) every 5–10 min and glucose infusion adjusted accordingly. Steady state was reached after 70 min and blood samples were collected on blood spot cards at 10-min intervals over 30 min to determine steady-state enrichment levels of 6,6-D2 D-glucose. Blood samples for serum were collected at the end of the basal infusion phase and at the end of the

hyperinsulinaemic phase for determination of FFAs and insulin levels during the clamp. Mice were then killed by cervical dislocation and the organs removed and frozen. Blood spots cards were extracted in 90% ethanol. Extracts underwent aldonitrile-pentaacetate derivatisation and mass spectral analysis as described in Supplementary Information. Insulin levels were determined by ELISA (Ultra Sensitive Mouse Insulin ELISA Kit, Crystal Chem) and FFA levels were determined enzymatically (HR Series NEFA-HR(2), WACO)

**Tissue-specific glucose uptake.** Tissue-specific glucose uptake was determined by intraperitoneal administration of 2-[$^3$H]-deoxyglucose (DG). Overnight fasted mice were injected with 1 g per kg D-glucose spiked with 2-[$^3$H]-DG (12 MBq [$^3$H]-DG (Perkin Elmer) per gram of D-glucose). Blood samples were collected from tail vein at 0, 15, 30 and 60 min following glucose injection. At the end of the time course, tissues were excised and snap frozen in liquid nitrogen. The 2-[$^3$H]-DG uptake into tissues was determined with the Ba(OH)$_2$/ZnSO$_4$ precipitation method. Briefly, 100 mg of tissue samples were homogenised in 1.5 ml (or 5 μl of plasma were diluted in 0.5 ml) of 0.5% perchloric acid. Lysates were clarified by centrifugation in a benchtop microcentrifuge. One millilitre of the clarified supernatant was neutralised by addition of 50 ml each of 10% KOH and 1 M potassium phosphate buffer pH 7.0. 0.55 ml of the neutralised supernatant was transferred to a scintillation vial (designated n). The remainder 0.55 ml was precipitated by addition of 100 ml each of 0.3 N Ba(OH)$_2$ and 0.3 N ZnSO$_4$ and centrifugation at 16,000×$g$ in a benchtop microcentrifuge. Totally, 0.6 ml of the supernatant was transferred to a scintillation vial (designated p). Disintegrations per minute (dpm) were measured in n and p aliquots by liquid scintillation counting. The difference in dpm between n and p represents uptake of 2-[$^3$H]-DG. Clearance of 2-[$^3$H]-DG and the metabolic index ($R_g$), as a measure of tissue-specific glucose uptake, were calculated using the following equations:

$$K_g = 2-[^3H]-_{tissue}/AUC\,2-[^3H]-DG_{plasma},$$

$$R_g = K_g \times Glucose_{plasma},$$

where 2-[$^3$H]-DGP$_{tissue}$ is the 2-[$^3$H]-DG-6-Phosphate radioactivity in the tissue in dpm per g of tissue weight, AUC 2-[$^3$H]-DG$_{plasma}$ is the area under the plasma 2-[$^3$H]-DG disappearance curve in dpm per ml per min and Glucose$_{plasma}$ is the average blood glucose concentration in millimolar during the glucose excursion[49].

**Protein extraction and immunoblot analysis.** Mouse tissues were harvested and flash frozen in liquid nitrogen and then stored at −80 °C until further processing. Samples were homogenised in a lysis buffer containing 50 mM Tris-HCl pH 7.4, 100 mM NaCl, 50 mM NaF, 5 mM EDTA, 2 mM EGTA, 40 mM beta-glycer-ophosphate, 10 mM sodium pyrophosphate and 1% Triton X-100 and protease inhibitors. Proteins were separated by sodium dodecyl sulfate polyacrylamide gel electrophoresis and then transferred to PVDF membrane. A total of 5% non-fat milk was used to saturate the membrane prior to incubation with primary antibodies. p110α (cat. no. 4255, diluted 1:1000), p110β (cat. no. 3011, diluted 1:1000), pT308 Akt (cat. no. 2965, diluted 1:1000), pS473 (cat. no. 4060, diluted 1:1000), total Akt (cat. no. 9272, diluted 1:1,000), pS660 HSL (cat. no. 4126, diluted 1:1000), total HSL (cat. no. 4107, diluted 1:1000), phospho-PKA substrate (cat. no. 9624, diluted 1:1000), total perilipin (cat. no. 9349, diluted 1:1000), pS133 CREB (cat. no. 9191, diluted 1:1000), p T180/Y182 p38 MAPK (cat. no. 9215, diluted 1:1000) and total p38 MAPK (cat. no 9212, diluted 1:1000) antibodies were from Cell Signaling Technology (CST). UCP1 antibody was from Abcam (cat. no. 10,983, diluted 1:1000). p85 antibody (cat. no. 06-195, diluted 1:1000) was from Millipore. Vinculin antibody (cat. no. V9264, diluted 1: 10,000) was from Sigma. PDE3b antibody (cat. no. STJ110746, diluted 1:1000) was from St. John's Laboratory. Membranes were then washed and incubated with HRP-conjugated, goat anti-Rabbit IgG (DAKO, cat. no. P0448, diluted 1:2000) and developed using ECL reagent (GE Healthcare) for radiographic film detection of p110α. Band intensity was quantified by ImageJ software. In all other cases, detection was performed with fluorescently labelled secondary antibodies (anti-mouse DyLight 800-conjugated, Rockland, cat. no. 610-145-002, diluted 1:5000 and anti-rabbit Alexa-Fluor 680-conjugated, Invitrogen, cat. no. A21076, diluted 1:5000), using an Odyssey CLx infra-red scanner (LICOR). Detection and quantification were performed using the manufacturer's software (Image Studio). Uncropped images of the most important blots are shown in Supplementary Information.

**Immunoprecipitation and PI3K assay.** For immunoprecipitation and PI3K lipid kinase assay, tissue lysates were prepared as described above. Immunoprecipitation was performed by mixing lysates (2 mg of protein) with 2 μg IRS-1 antibody (Millipore, cat. no. 05-1085). Following overnight incubation at 4 °C, immune complexes were captured by addition of protein-A-Sepharose beads (GE Healthcare) and incubation for an additional 1 h. Bead-associated immune complexes were subjected to lipid kinase assays. Lipid kinase assays were performed in a total volume of 50 μl in a buffer containing 50 mM HEPES (pH 7.4), 100 mM NaCl, 1 mM dithiothreitol, 5 mM MgCl$_2$, 100 μM ATP (plus 0.1 μCi of [γ-$^{32}$P]-ATP per assay) using 200 μg ml$^{-1}$ phosphatidylinositol (Sigma) as a substrate. Reactions were incubated at 25 °C for 20 min and terminated by the addition of 100 μl of 0.1 M HCl and 200 μl of chloroform:methanol (1:1).

The mixture was vortexed and the phases were separated by centrifugation at $10,000 \times g$ for 2 min. The lower organic phase was spotted onto thin layer silica (TLC) gel-60 plates (Merck), which had been treated with 1% oxalic acid, 1 mM EDTA in water:methanol (6:4). TLC plates were developed in a solvent consisting of propanol-1:2 M acetic acid (13:7). Images of radiolabelled lipid products were captured using a Fuji FLA-2000 phosphorimager and analysed using ImageJ software.

**UCP1 immunohistochemistry**. Adipose tissues were fixed in formalin for 24 h and embedded in paraffin. Tissue sections were deparaffinised, subjected to antigen unmasking (boiling in 10 mM sodium citrate pH 6 in a pressure cooker for 30 min), blocked with 5% bovine serum albumin (BSA) and incubated with UCP1 antibody (Abcam, cat. no. 10983) diluted 1:1000 for 16 h at 4 °C. Sections were then incubated with SignalStain© Boost IHC Detection Reagent (Cell Signaling Technology, cat. no. 8114P) and immune complexes were visualised with SignalStain© DAB reagent (Cell Signaling Technology, cat. no. 8059).

**RNA extraction and qPCR**. RNA was extracted from adipose tissue samples using Triazol (Thermo Fisher) and reverse transcribed to cDNA using Qiagen Omniscript RT kit according to the manufacturer's instructions. Quantitative PCR (qPCR) was performed using an ABI7900HT Fast Real Time PCR detection system and Power SYBR-Green PCR Master Mix (ABI). Relative gene expression was quantified using the $2^{-\Delta\Delta Ct}$ method. Mouse ribosomal protein S18 (rps18) was used as a loading control. A list of primers used in the present study is given in Supplementary Table 1.

**Lipolysis and cAMP levels in adipose tissue explants**. Adipose tissue explants were adjusted to approximately 100 mg of weight and then minced with fine scissors. Explants were incubated in serum-free DMEM supplemented with 0.2% (w/v) fatty acid-free BSA for 2 h at 37 °C in 5% $CO_2$ incubator then transferred to Kreb's–Riger's buffer (30 mM Hepes pH 7.4, 10 mM NaHCO_3, 120 mM NaCl, 4 mM $KH_2PO_4$, 1 mM $MgSO_4$, 0.75 mM $CaCl_2$) supplemented with 0.2% fatty acid-free BSA and 10 mM glucose and incubated for an additional hour. Inhibitors were added for the last 15 min of the incubation. Lipolysis was stimulated by addition of 1 µM Isoproterenol or Norepinephrine-L-bitartate (Sigma). Totally, 50 µl of medium was removed 30 min following stimulation and glycerol content was determined using Glycerol Reagent and Glycerol Standard (Sigma). For determination of intracellular cAMP levels tissue explants were processed, incubated and stimulated as above. Explants were collected 20 min following adrenergic stimulation, frozen in dry ice and kept in −80 °C until further processing. Samples were homogenised in 0.5 ml of 0.1 N HCl using a mechanical homogeniser. Homogenates were clarified by centrifugation for 5 min in a benchtop refrigerated microcentrifuge. Supernatants were diluted 1 in 4 with cAMP ELISA buffer and the cAMP content was determined with a cyclic AMP Select ELISA kit (Cayman, cat. no. 501040).

**Phosphodiesterase activity assay**. BAT-derived adipocytes were lysed in a buffer containing 20 mM Hepes-NaOH (pH 7.4), 0.5 mM EDTA, 2 mM MgCl₂, 0.1% Triton X-100, 0.5 mM DTT, 1 mM EGTA, 1 µM microcystin-LR and protease inhibitors. Lysates were briefly sonicated using a probe sonicator and were desalted using Zeba Spin desalting columns, 7 kD molecular weight cut-off (Thermo Scientific, cat. no. 89882), equilibrated in PDE assay buffer. Desalted lysates (15 µg protein) were assayed for phosphodiesterase activity using a kit from Abcam (cat. no. ab139460) according to the manufacturer's instructions.

**Indirect calorimetry**. EE and norepinephrine-induced thermogenesis were measured using a Metatrace gas analyser (Creative Scientific, UK). The gas analyser uses a paramagnetic oxygen analyser and an infra-red $CO_2$ sensor to measure oxygen and carbon dioxide concentrations. The analyser operates in push mode with airflow to each chamber controlled by a mass flow controller set to flow 400 ml per min. Cages were multiplexed to the analyser, with four cages being read by each analyser. For free-living EE cages were sampled for 30 s every 11 min, with a 90 s washout period between each chamber. EE was calculated from the $VO_2$ and $CO_2$ according to the modified Weir equation: EE $(J\,min^{-1}) = 15.818 \times VO_2$ $(ml\,min^{-1}) + 5.176 \times VCO_2$ $(ml\,min^{-1})$. For NE-induced thermogenesis, mice were anesthetized with sodium pentobarbital $(90\,mg\,kg^{-1})$ and placed in a 30 °C chamber. Mice were injected with 0.5 mg NE per kg. The dose of NE gave an average increase in EE of ~40% over the basal EE, while normal maximal thermogenic capacity experiments were for mice acclimated to a similar temperature a 160% increase would be expected[50]. Therefore, the applied NE dose fulfilled the requirement of being a sub-maximal dose and could potentially allow detection of changes in sensitivity to NE, not just changes in brown adipose tissue thermogenic capacity.

**Statistical analysis**. Data are presented as mean ± standard error of the mean (SEM). Statistical analyses were performed with GraphPad Prism 8. Specific tests are detailed in the figure legends.

**Study approval**. All experimental procedures complied with the UK Home Office Animals (Scientific Procedures) Act 1986 and were performed with the approval of the UCL Animal Welfare and Ethical Review Body.

**Reporting summary**. Further information on experimental design is available in the Nature Research Reporting Summary linked to this article.

## Data availability
The data that support the findings of this study are included in this published article and its Supplementary Information files. All other relevant data are available from the corresponding author upon reasonable request.

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

## Acknowledgements

This study was funded by a Wellcome Trust University Award (093115/Z/10/Z) awarded to L.F. Calorimetry and hyperinsulinemic–euglycemic clamp experiments were conducted at the Wellcome Trust-MRC IMS Disease Model Core funded by the MRC Metabolic Diseases Unit (MRC_MC_UU_12012/5) and Wellcome Trust Strategic Award (100574/Z/12/Z) and supported by a BHF Programme grant (RG/18/7/33636). The authors wish to thank Evan Rosen, Beth Israel Deaconess Medical Center, for providing Adipoq–Cre mice.

## Author contributions

L.C.F. conceived the study and drafted the paper; C.A., A.Y., L.B., I.S., S.V., A.V.P. and L.C.F. designed the research; C.A., A.Y., L.B., I.S., S.V., A.K.M. and L.C.F. acquired and analysed data; C.A., A.Y., L.B., I.S., S.V., C.D., A.V.P. and L.C.F. interpreted the data and edited the paper.

## Additional information

**Competing interests:** The authors declare no competing interests.

