## [Peer Review File · Nature Communications]

Reviewers' comments:

Reviewer #1 (Remarks to the Author):

This study investigates the role of p110alpha in adipose tissue. The results indicate that the beneficial metabolic effect of p110alpha inactivation is due to a potentiating effect on beta-adrenergic signaling. The results are clearly presented, novel and of interest to the field, and the manuscript reads very well. However, there are several issues that must be clarified.

General comments

Figure 2. AUCs are recommended for ITTs, specially because at 18 month-old it is not clear that the conditional KOs are "profoundly insulin resistant". At this particular age, they do not maintain normal glucose tolerance.

Figure 5. The n used in the different panels is too low (n=2-3). Admittedly this reviewer is not a statistical expert but I am very surprised that with such a low number of animals per group the authors found statistical differences. This comment may also apply to figure 4, where n is 3-4.

Figure 6. Results concerning the browning of WAT are not entirely convincing. First, the response of the mice to beta-3 adrenoreceptor agonist seems to be clear (in terms of RNA expression) but how is the situation in WAT of control versus conditional KOs in basal conditions? And in mice fed a HFD? According to data on energy expenditure, conditional KOs are more sensitive to the stimulation of beta-3-AR, but this does not mean that the lack of p110alpha directly affects BAT activity.

Second, to justify browning just with the measurement of UCP1 and CIDEA is somehow weak, especially because there are serious doubts about the correlation between UCP1 gene expression and its thermogenic capacity. Additional approaches should be used to show the browning of WAT.

Figure 7. How do the authors explain the different body weight between genotypes? If there are no differences in food intake, respiratory exchange ratio or energy expenditure, this is difficult to understand. BAT of conditional KOs is using much more glucose, and according to that one could assume that the thermogenic activity of BAT is increased, but this does not seem to be the case...perhaps the lack of p110alpha in adipose tissue affects to adipocyte differentiation? This aspect deserves further investigation.

Reviewer #2 (Remarks to the Author):

- The in vivo glucose uptake is increased only in iBAT of p110alpha DEL mice, whereas no changes were observed in the other analysed tissues. This outcome was used by the authors to explain the better glucose tolerance displayed by the p110alpha DEL mice. However, the experiments with tissue explants did not show any potentiating effect of p110alpha DEL on Beta-adrenergic stimulation of iBAT, but only in sqWAT, where no changes in glucose uptake were verified. How does the authors explain such conflicting data?

- The authors made a statement that p110alpha DEL mice displays browning of the sqWAT. This is a quite incorrect statement, given that in the absence of the beta-adrenergic agonist CL stimulation there is no browning. What the authors did see was that the p110alpha DEL mice were more sensitive to the CL effect on UCP1 mRNA expression in sqWAT. In addition to mRNA changes, browning also implies the increase in UCP1 protein expression, presence of multilocular lipids in the cytoplasm and abundance of mitochondria due to the higher mitochondrial biogenesis. It is likely that the authors did not see this phenotype in the p110alpha DEL mice with no stimuli.

- The data on energy expenditure in response to acute injections of CL suggest p110alpha DEL mice are more sensitive to CL injections. However, if that is true, it must be due to the potentiating effect on iBAT rather than sqWAT, since the acute effects of CL on energy expenditure are exclusively due to its effect on iBAT. This concept is also in disagreement with the authors statements that p110alpha deletion in fat improves thermogenesis due to the potentiating effect on sqWAT rather than iBAT. sqWAT contribution to thermogenesis is not relevant when acutely stimulated by cold or agonist.

- If the Beta-3 adrenergic receptor is upregulated in iBAT of p110alpha DEL mice, it is not clear why the potentiating effect is present only in sqWAT.

Reviewer #3 (Remarks to the Author):

The manuscript provides the first genetic evidence showing that reductions in p110alpha isoform specifically in adipose tissue have beneficial effects on whole body glucose metabolism. Importantly the study deletes PIK3CA with Adipoq CRE and so achieves a more adipose selective deletion than would be achieved by other promoters such as AP2 Cre. In general their findings support those seen in mice in which the insulin receptor is selectively deleted in adipose tissues. These authors (and several previous studies) have also shown that small molecule inhibitors of p110alpha bring about broadly similar changes in glucose metabolism. The main claim to novelty of the current study is the

proposal that the p110alpha inhibition is reducing the inhibitory effects that insulin normally has on cAMP levels in adipose tissue. Thus they propose the p110 alpha deletion is potentiating the effects of signals through metabolic regulatory pathways such as the adrenergic signalling system. The contention is that this would act through the regulation of PDE3B activity via Akt mediated phosphorylation. Indeed there is strong evidence in the literature that such phosphorylation can indeed regulate PDE3B activity (PMID9421418) and loss of PDE3B in white adipocytes leads to browning (PMID28084425) but the authors of this manuscript do not directly measure PDE3B phosphorylation or PDE3B levels or activity which is a shortcoming. Of interest beta adrenergic signalling is known to down regulate PDE3B while insulin upregulates it (PMID17198676) so changes in PDE3B levels are a possible explanation. Having said that the evidence in support the mechanism proposed rests with surrogate markers of PDE activity and the results presented may not be entirely consistent with the hypothesis. For example, when the system is pushed fully with ISO there is a only small additional effect on glycerol release and cAMP production with A66 or p110alpha deletion in WAT (Fig 4c,d) but this really isn't clear in the presence of the physiological agonist NE (suppl data). Further, the phospho-CREB response in the DEL mice only seems to be the same as FLOX mice in response to ISO (Fig. 4a,b). In reality, the effects observed could perhaps better be explained by the increase in levels of the beta-3 adrenergic receptor in brown fat and the consequent effects on brown fat. It will be important to know whether there is any further data on differential effects of p110alpha deletion on gene expression patterns in different types of tissue.

There are several other issues which are not fully addressed in the manuscript.

1. Fasting insulin is an important marker of effects on whole body metabolism and it is striking to see that there is a sex specific dimorphism in the impact to p110alpha deletion on this parameter. Based on this there needs to be more discussion of potential reasons for these differences and the implications of this.
2. While the deletion of p110alpha is selectively in adipocytes insulin signalling is also attenuated in liver and muscle. No convincing explanation for this is provided. Also, what is the status of insulin signalling in BAT ?
3. The reductions in leptin and adiponectin with p110alpha deletion are large and could potentially be contributing to the effects seen. How were these changes controlled for ?
4. Long term treatment with inhibitors of p110alpha has been shown to reduce movement of animals (PMID 23837532). This may well be due to effects of PI3K inhibitors in the brain so it is probably not going to be seen in the p110 alpha DEL mice in this study but it would be interesting to know if the investigators have data on this.
5. The authors suggest a mechanism by which adrenergic signalling might be signalling through PI3K to pAkt but the data in Fig5 doesn't really support that as the level of phosphorylation induced by ISO alone is still seen in the presence of A66 so the more likely explanation is cAMP dependant pathway e.g. an EPAC mediated mechanism (PMID 16839743).

Response to Referee comments

Reviewer #1 (Remarks to the Author):

This study investigates the role of p110alpha in adipose tissue. The results indicate that the beneficial metabolic effect of p110alpha inactivation is due to a potentiating effect on beta-adrenergic signaling. The results are clearly presented, novel and of interest to the field, and the manuscript reads very well. However, there are several issues that must be clarified.

General comments

-Figure 2. AUCs are recommended for ITTs, specially because at 18 month-old it is not clear that the conditional KOs are “profoundly insulin resistant”. At this particular age, they do not maintain normal glucose tolerance.

Authors' response: AUCs for the ITTs are now presented in Fig. 2. The difference between the two genotypes is not statistically significant at 18 months, but this is likely due to variability in the response to insulin injections and in glucose measurements which is not uncommon in standard ITTs and GTTs. Given that the 24 month time point shows insulin resistance, it seems probable that they were also insulin resistant at 18 months. Our confidence in this concept is due to the clamps we performed at 20 weeks where we detected a reduction in GIR of over 60% with a P-Value of less than 10E-6. Therefore we know the mice are resistant at 5 months and 24 months. The statement 'profoundly insulin resistant' refers to this data from the insulin clamp, which is the most accurate method of quantification of insulin sensitivity. We have amended the relevant section of the manuscript to precisely reflect this point. Indeed, at this particular age point (18 months), the mutant male mice appear to be slightly more glucose tolerant than their control counterparts.

-Figure 5. The n used in the different panels is too low (n=2-3). Admittedly this reviewer is not a statistical expert but I am very surprised that with such a low number of animals per group the authors found statistical differences.

Authors' response: We have now performed quantification of beta3-AR protein in iWATs and BATs of additional mice (n=8-9 per genotype in total) from eight different litters and two different generations and it turns out that the difference that we originally observed using 3 mice per genotype from the same litter was due to a batch effect and that the difference in beta3-AR expression between p110 α ^{FLOX} and p110 α ^{DEL} mice is only marginal. We have now amended the respective text and the figure (data now shown in Suppl. Fig. 7).

-This comment may also apply to figure 4, where n is 3-4.

Authors' response: We present data from experiments in explants of 4 mice per genotype, which were performed under exactly the same conditions and have therefore been pooled together for the statistical analysis shown in Fig. 4a. Of course, we have done many more experiments in tissue explants over the years all showing the same effect. The same effect we have shown in human MADS derived adipocytes (Fig. 4e) and now we have added data showing the same effect in BAT

explants (Fig. 5c) and in BAT-derived immortalised adipocytes (Fig. 5b). Therefore, we believe that taken altogether these data provide sufficient evidence for the proposed mechanism.

-Figure 6. Results concerning the browning of WAT are not entirely convincing. First, the response of the mice to beta-3 adrenoreceptor agonist seems to be clear (in terms of RNA expression) but how is the situation in WAT of control versus conditional KOs in basal conditions? And in mice fed a HFD?

Authors' response: We did not consistently detect increased expression of UCP-1 protein in iWAT of p110 α ^{DEL} mice at the basal level. It is only upon stimulation with beta3-AR agonist that we detect higher levels of UCP-1 protein in iWATs of p110 α ^{DEL} mice compared with p110 α ^{FLOX} mice. However, we do detect increased expression of UCP-1 in the BAT of p110 α ^{DEL} mice at the basal level (we used BAT from 8 mice per genotype, originating from 8 different litters and 3 different generations). In terms of treatment with HFD diet, in line with previous reports, we found that short term HFD feeding increases UCP-1 expression in BAT with p110 α ^{DEL} mice still expressing higher levels of UCP-1. These data are shown in Fig. 3e-f and Suppl. Fig. 5.

-According to data on energy expenditure, conditional KOs are more sensitive to the stimulation of beta-3-AR, but this does not mean that the lack of p110alpha directly affects BAT activity. Second, to justify browning just with the measurement of UCP1 and CIDEA is somehow weak, especially because there are serious doubts about the correlation between UCP1 gene expression and its thermogenic capacity. Additional approaches should be used to show the browning of WAT.

Authors' response: Indeed, mRNA expression of UCP-1 does not necessarily correlate with protein levels. We have now added data for additional thermogenic markers (PGC-1a, Dio2, elov13) as well as western blot and immunohistochemical detection of UCP-1 protein (all shown in Fig. 6).

-Figure 7. How do the authors explain the different body weight between genotypes? If there are no differences in food intake, respiratory exchange ratio or energy expenditure, this is difficult to understand. BAT of conditional KOs is using much more glucose, and according to that one could assume that the thermogenic activity of BAT is increased, but this does not seem to be the case...perhaps the lack of p110alpha in adipose tissue affects to adipocyte differentiation? This aspect deserves further investigation.

Authors' response: Differences in energy expenditure that would account for the observed weight differences would be extremely difficult to detect by calorimetry in this case. We performed calorimetry in approx. 1 year old mice. In the period between 5 and 13 months of age the mice only diverge by ~5 grams, that is ~25 mg per day. A mouse expends 45 kJ/day which is 1.25 g of fat and consumes about 3.3 g or 45 kJ of food. So the measurements would need to be a) in the correct time period to detect a divergence b) have the ability to detect a 1-2% difference in energy expenditure or food intake. In the calorimetry study we performed our mice exhibited a coefficient of variation of 8% in terms of their metabolic rate, enabling us to reliably detect an 11% difference in means with a power of 0.8. To detect a 2% difference in metabolic rate, we would have needed approximately **256** mice.

Finally, an important point is that altering BAT function does not necessarily increase free living metabolic rate – UCP1 KO mice, which have no BAT thermogenic capacity, have normal energy expenditure at room temperature (PMID: 16914547) and tend to be lean at room temperature (PMID: 9139827). The energy expenditure in measurements in response to norepinephrine that we

conducted in Figure 6i is considered a better readout of changes in brown fat function (PMID: 11511509)

With regards to adipogenesis, we have seen no evidence to suggest an effect of p110 α inhibition on adipocyte differentiation. Adipocytes from p110 α ^{DEL} mice look morphologically normal and express normal levels of HSL, perilipin and adrenergic receptors (and they are more responsive to adrenergic stimulation of the lipolytic cascade). Furthermore, BAT-derived adipocytes differentiate normally *ex vivo*.

Reviewer #2 (Remarks to the Author):

- The in vivo glucose uptake is increased only in iBAT of p110alpha DEL mice, whereas no changes were observed in the other analysed tissues. This outcome was used by the authors to explain the better glucose tolerance displayed by the p110alpha DEL mice. However, the experiments with tissue explants did not show any potentiating effect of p110alpha DEL on Beta-adrenergic stimulation of iBAT, but only in sqWAT, where no changes in glucose uptake were verified. How does the authors explain such conflicting data?

Authors' response: The original version of the manuscript did not include signalling data from BAT. We have now added data from BAT explants and from immortalised brown adipocytes, showing potentiation of adrenergic signalling in BAT as well. These data are shown in Fig. 5 b-c.

- The authors made a statement that p110alpha DEL mice displays browning of the sqWAT. This is a quite incorrect statement, given that in the absence of the beta-adrenergic agonist CL stimulation there is no browning. What the authors did see was that the p110alpha DEL mice were more sensitive to the CL effect on UCP1 mRNA expression in sqWAT. In addition to mRNA changes, browning also implies the increase in UCP1 protein expression, presence of multilocular lipids in the cytoplasm and abundance of mitochondria due to the higher mitochondrial biogenesis. It is likely that the authors did not see this phenotype in the p110alpha DEL mice with no stimuli.

Authors' response: This is a valid point. We did not consistently observe browning of subcutaneous WAT in the absence of stimulation with adrenergic agonist. We have amended the text in the relevant sections to precisely reflect this point.

- The data on energy expenditure in response to acute injections of CL suggest p110alpha DEL mice are more sensitive to CL injections. However, if that is true, it must be due to the potentiating effect on iBAT rather than sqWAT, since the acute effects of CL on energy expenditure are exclusively due to its effect on iBAT. This concept is also in disagreement with the authors statements that p110alpha deletion in fat improves thermogenesis due to the potentiating effect on sqWAT rather than iBAT. sqWAT contribution to thermogenesis is not relevant when acutely stimulated by cold or agonist.

Authors' response: As mentioned above (response to first comment) inhibition of p110 α potentiates adrenergic signalling in BAT as well and as the Referee points out this provides a plausible mechanism underlying the reported phenotypes.

- If the Beta-3 adrenergic receptor is upregulated in iBAT of p110alpha DEL mice, it is not clear why the potentiating effect is present only in sqWAT.

Authors' response: As mentioned above (response to second comment of Referee 1), we have re-evaluated the expression levels of beta3-AR and found that the upregulation we reported in our original version was due to a batch effect. The potentiating effect of p110 α inhibition is also evident in BAT.

Reviewer #3 (Remarks to the Author):

-The manuscript provides the first genetic evidence showing that reductions in p110alpha isoform specifically in adipose tissue have beneficial effects on whole body glucose metabolism. Importantly the study deletes PIK3CA with Adipoq CRE and so achieves a more adipose selective deletion than would be achieved by other promoters such as AP2 Cre. In general their findings support those seen in mice in which the insulin receptor is selectively deleted in adipose tissues. These authors (and several previous studies) have also shown that small molecule inhibitors of p110alpha bring about broadly similar changes in glucose metabolism. The main claim to novelty of the current study is the proposal that the p110alpha inhibition is reducing the inhibitory effects that insulin normally has on cAMP levels in adipose tissue. Thus they propose the p110 alpha deletion is potentiating the effects of signals through metabolic regulatory pathways such as the adrenergic signalling system. The contention is that this would act through the regulation of PDE3B activity via Akt mediated phosphorylation. Indeed there is strong evidence in the literature that such phosphorylation can indeed regulate PDE3B activity (PMID9421418) and loss of PDE3B in white adipocytes leads to browning (PMID28084425) but the authors of this manuscript do not directly measure PDE3B phosphorylation or PDE3B levels or activity which is a shortcoming. Of interest beta adrenergic signalling is known to down regulate PDE3B while insulin upregulates it (PMID17198676) so changes in PDE3B levels are a possible explanation. Having said that the evidence in support the mechanism proposed rests with surrogate markers of PDE activity and the results presented may not be entirely consistent with the hypothesis.

Authors' response: The literature quoted by the Referee above provides important evidence that we had missed in the original version and we have now cited in the revised version of the manuscript as our data are in line with those previously reported findings. In addition to Akt phosphorylation we have now tested PDE activity in extracts of BAT-derived adipocytes and found that adrenergic stimulation induces a PI3K-dependent increase in PDE activity. These data are shown in Fig. 5d-e. We have also quantified PDE3b protein levels in iWAT extracts and found them only marginally lower in p110 α ^{DEL} iWATs (Suppl. Fig. 8).

-For example, when the system is pushed fully with ISO there is a only small additional effect on glycerol release and cAMP production with A66 or p110alpha deletion in WAT (Fig 4c,d) but this really isn't clear in the presence of the physiological agonist NE (suppl data). Further, the phospho-CREB response in the DEL mice only seems to be the same as FLOX mice in response to ISO (Fig. 4a,b). In reality, the effects observed could perhaps better be explained by the increase in levels of the beta-

3 adrenergic receptor in brown fat and the consequent effects on brown fat. It will be important to know whether there is any further data on differential effects of p110alpha deletion on gene expression patterns in different types of tissue.

Authors' response: We agree that activation of BAT rather than browning in iWAT more likely underlies the phenotypic effects documented in our study. The data showing enhanced basal and diet-induced activation of BAT (Fig. 3e-f and Suppl. Fig. 5) in the revised manuscript add further support to this notion. The data from the experiments in iWAT support the proposed mechanism of potentiation of adrenergic signalling through p110 α inhibition rather than highlight a key role of this depot in the gross phenotypic effects.

We do not have high-throughput gene expression data from various adipose tissues, as we felt that a study of gene expression at such a large scale (two genotypes-three fat tissue types-two sexes-a number of individual mice per sex/ genotype/tissue type) was beyond the scope of this investigation.

-There are several other issues which are not fully addressed in the manuscript.

1. Fasting insulin is an important marker of effects on whole body metabolism and it is striking to see that there is a sex specific dimorphism in the impact to p110alpha deletion on this parameter. Based on this there needs to be more discussion of potential reasons for these differences and the implications of this.

Authors' response: Sexual dimorphism is not uncommon in metabolic phenotypes. Especially when adipose tissue is involved, the function of which is well-known to be influenced by sex in both mice and humans. Such effects could account for a differential impact of p110 α inactivation on insulin sensitivity and thus on fasting insulin levels between the two sexes at the age point of this measurement. ITTs also showed that female p110 α ^{DEL} mice were substantially more insulin resistant than their control littermates, in contrast to male mice, at the age of 12 months (Suppl. Fig. 3). We have now added a statement in the respective section of the revised manuscript.

-2. While the deletion of p110alpha is selectively in adipocytes insulin signalling is also attenuated in liver and muscle. No convincing explanation for this is provided. Also, what is the status of insulin signalling in BAT ?

Authors' response: It is not unusual that interventions affecting insulin sensitivity in one organ cause systemic effects by affecting other metabolic organs. This is especially true for an organ such as the adipose tissue that produces adipokines which affect other metabolic tissues. An example is adipose tissue-specific deletion of GLUT4 which causes insulin resistance in liver and muscle (PMID: 11217863).

Insulin signalling is only marginally affected in BAT, likely due to a role of the PI3K p110 β isoform in this tissue which might compensate for loss of p110 α (these data are now shown in Fig. 1e).

-3. The reductions in leptin and adiponectin with p110alpha deletion are large and could potentially be contributing to the effects seen. How were these changes controlled for ?

Authors' response: This is a good point and it might relate to the development of insulin resistance in liver and muscle as mentioned above. We have also added a statement in the manuscript to acknowledge this possibility.

-4. Long term treatment with inhibitors of p110alpha has been shown to reduce movement of animals (PMID 23837532). This may well be due to effects of PI3K inhibitors in the brain so it is probably not going to be seen in the p110 alpha DEL mice in this study but it would be interesting to know if the investigators have data on this.

Authors' response: We measured activity during the calorimetry run and it was unchanged. As the Referee points out, it is unlikely that deletion of p110 α in adipose tissue would affect locomotor activity. The article quoted above reports prolonged administration of a cancer therapeutic dose (10 mg/kg, intraperitoneally) of A66. At such a dose, the inhibitor will most likely have off-target effects and multiple toxicities. Of note, it has been reported elsewhere that single oral administration (15 mg/kg) of three PI3K inhibitors did not affect locomotor activity in mice (PMID: 27816049, Fig. 4B).

-5. The authors suggest a mechanism by which adrenergic signalling might be signalling through PI3K to pAkt but the data in Fig5 doesn't really support that as the level of phosphorylation induced by ISO alone is still seen in the presence of A66 so the more likely explanation is cAMP dependant pathway e.g. an EPAC mediated mechanism (PMID 16839743).

Authors' response: Fig 5a shows that the PI3K inhibitor reduces the level of ISO-stimulated pAkt by approx. 70%, which is a very substantial inhibition. As mentioned in the revised manuscript, EPAC has been shown before to activate PI3K. We have now assessed the potential involvement of EPAC using the EPAC inhibitor ESI09 and we found that EPAC inhibition tends to inhibit NE-stimulated Akt phosphorylation and PDE activity, thus providing a molecular link between beta-AR stimulation and PI3K/Akt activation. These data are shown in Fig. 5d-e.

REVIEWERS' COMMENTS:

Reviewer #1 (Remarks to the Author):

The authors have addressed all my concerns and i have no further comments

Reviewer #3 (Remarks to the Author):

The authors have adequately addressed my concerns raised in my initial review.